# PUSHING THE ACCURACY-GROUP ROBUSTNESS FRONTIER WITH INTROSPECTIVE SELF-PLAY

**Jeremiah Zhe Liu, Krishnamurthy Dj Dvijotham, Jihyeon Lee, Quan Yuan,**
**Balaji Lakshminarayanan**[*]**, Deepak Ramachandran**[*]
Google Research
{jereliu,dvij,jihyeonlee,yquan, balajiln,ramachandrand}@google.com

## ABSTRACT

Standard empirical risk minimization (ERM) training can produce deep neural network (DNN) models that are accurate on average but underperform in underrepresented population subgroups, especially when there are imbalanced group distributions in the long-tailed training data. Therefore, approaches that improve the accuracy - group robustness tradeoff frontier of a DNN model (i.e. improving worst-group accuracy without sacrificing average accuracy, or vice versa) is of crucial importance. Uncertainty-based active learning (AL) can potentially improve the frontier by preferentially sampling underrepresented subgroups to create a more balanced training dataset. However, the quality of uncertainty estimates from modern DNNs tend to degrade in the presence of spurious correlations and dataset bias, compromising the effectiveness of AL for sampling tail groups. In this work, we propose *Introspective Self-play (*ISP*)*, a simple approach to improve the uncertainty estimation of a deep neural network under dataset bias, by adding an auxiliary *introspection* task requiring a model to predict the bias for each data point in addition to the label. We show that ISP provably improves the *bias-awareness* of the model representation and the resulting uncertainty estimates. On two real-world tabular and language tasks, ISP serves as a simple "plug-in" for AL model training, consistently improving both the tail-group sampling rate and the final accuracy-fairness trade-off frontier of popular AL methods.

## 1 INTRODUCTION

Modern deep neural network (DNN) models are commonly trained on large-scale datasets (Deng et al., 2009; Raffel et al., 2020). These datasets often exhibit an imbalanced long-tail distribution with many small population subgroups, reflecting the nature of the physical and social processes generating the data distribution (Zhu et al., 2014; Feldman & Zhang, 2020). This imbalance in training data distribution, i.e., **dataset bias**, prevents deep neural network (DNN) models from generalizing equitably to the underrepresented population groups (Hasnain-Wynia et al., 2007).

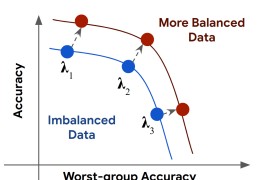

Figure 1: Example of accuracy-fairness frontier. Under a more balanced training data distribution, the model can attain a better accuracy-fairness frontier (**Red**) when compared to training under an imbalanced distribution (**Blue**) at every tradeoff level $\lambda$ (Equation (1)).

**Accuracy-Group Robustness Frontier:** In response, the existing bias mitigation literature has focused on improving training procedures under a fixed and imbalanced training dataset, striving to balance performance between model accuracy and fairness (e.g., the average-case v.s. worst-group performance) (Agarwal et al., 2018; Martinez et al., 2020; 2021). Formally, this goal corresponds to identifying an optimal model $f \in \mathscr{F}$ that attains the *Pareto efficiency frontier* of the accuracy-group robustness trade-off (e.g., see Figure 1), so that under the same training data $D = \{y_i, \mathbf{x}_i\}_{i=1}^n$, we cannot find another model $f' \in \mathscr{F}$ that outperforms $f$ in both accuracy and worst-group performance. In the literature, this *accuracy-group robustness frontier* is often characterized by a trade-off objective (Martinez et al., 2021):

$$f_\lambda = \underset{f \in \mathscr{F}}{\arg\min} \, F_\lambda(f|D); \qquad F_\lambda(f|D) := R_{acc}(f|D) + \lambda R_{robust}(f|D), \tag{1}$$

where $R_{acc}$ and $R_{robust}$ are risk functions for a model's accuracy and group robustness (modeled here-in as worst-group accuracy), and $\lambda > 0$ a trade-off parameter. Then, $f_\lambda$ cannot be outperformed by any other $f'$ at the same trade-off level $\lambda$. The entire frontier under a dataset $D$ can then be

---

[*]Co-senior authors.

characterized by finding $f_\lambda$ that minimizes the robustness-accuracy objective (1) at every trade-off level $\lambda$, and tracing out its $(R_{acc}, R_{robust})$ performances (Figure 1).

**Goal:** However, the limited size of the tail-group examples restricts the DNN model's worst-group performance, leading to a compromised accuracy-group robustness frontier (Zhao & Gordon, 2019; Dutta et al., 2020), and thus we ask: *Under a fixed learning algorithm, can we meaningfully push the model's accuracy-group robustness frontier by improving the training data distribution using active learning?* That is, denoting by $D_{\alpha,n} = \{(y_i, \mathbf{x}_i)\}_{i=1}^n$ a training dataset with $K$ subgroups and the group size distribution $\alpha = [\alpha_1, \ldots, \alpha_K]$, we study whether a model's accuracy-group robustness performance $F_\lambda$ can be improved by rebalancing the group distribution of the training data $D_{\alpha,n}$, i.e., we seek to optimize an outer problem:

$$\underset{\alpha \in \Delta^{|\mathcal{G}|}}{\text{minimize}} \left[ \min_{f \in \mathcal{F}} F_\lambda(f | D_{\alpha,n}) \right], \tag{2}$$

where $\Delta^K$ is the simplex of all possible group distributions (Rolf et al., 2021). Our key observation is that given a sampling model with *well-calibrated* uncertainty (i.e., the model uncertainty is well-correlated with generalization error), active learning (AL) can preferentially acquire tail-group examples from unlabelled data *without needing group annotations*, and add them to the training data to reach a more balanced data distribution (Branchaud-Charron et al., 2021). Appendix A.5 discusses the connection between group robustness with fairness.

**Challenges with DNN Uncertainty under Bias:** However, recent work suggests that a DNN model's uncertainty estimate is less trustworthy under spurious correlations and distributional shift, potentially compromising the AL performance under dataset bias. For example, Ovadia et al. (2019) show that a DNN's expected calibration error increases as the testing data distribution deviates from the training data distribution, and Ming et al. (2022) show that a DNN's ability in detecting out-of-distribution examples is significantly hampered by spurious patterns. Looking deeper, Liu et al. (2022); Van Amersfoort et al. (2020) suggest that this failure mode in DNN uncertainty can be caused by an issue in representation learning known as *feature collapse*, where the DNN over-focuses on correlational features that help to distinguish between output classes on the training data, but ignore the non-predictive but semantically meaningful input features that are important for uncertainty quantification (Figure 2). In this work, we show that this failure mode can be provably mitigated by a training procedure we term *introspective training* (Section 2). Briefly, introspective training adds an auxiliary *introspection* task to model training, asking the model to predict whether an example belongs to an underrepresented group. It comes with a guarantee in injecting *bias-awareness* into model representation (Proposition 1), encouraging it to learn diverse hidden features that distinguish the minority-group examples from the majority, even if these features are not correlated with the training labels. Hence it can serve as a simple "plug-in" to the training procedure of any active learning method, leading to improved uncertainty quality for tail groups (Figure 2).

**Contributions:** In summary, our contributions are:

- We introduce ***Introspective Self-play*** (**ISP**), a simple training approach to improve a DNN model's uncertainty quality for underrepresented groups (Section 2). Using group annotations from the training data, ISP conducts *introspective training* to provably improve a DNN's representation and uncertainty quality for the tail groups. When group annotations are not available, ISP can be combined with a cross-validation-based *self-play* procedure that uses a noise-bias-variance decomposition of the model's generalization error (Domingos, 2000).
- **Theoretical Analysis.** We theoretically analyze the optimization problem in Equation (2) under a group-specific learning rate model (Rolf et al., 2021) (Section 3). Our result elucidates the dependence of the group distribution $\alpha$ in the model's best-attainable accuracy-group robustness frontier $F_\lambda$. In particular, it confirms the theoretical necessity of up-sampling the underrepresented groups for obtaining the optimal accuracy-group robustness frontier, and reveals that *underrepresentation* is in fact caused by an interplay of the subgroup's learning difficulty and its prevalence in the population.
- **Empirical Effectiveness.** Under two challenging real-world tasks (census income prediction and toxic comment detection), we empirically validate the effectiveness of ISP in improving the performance of AL with a DNN model under dataset bias (Section 4). For both classic and state-of-the-art uncertainty-based AL methods, ISP improves tail-group sampling rate, meaningfully pushing the accuracy-group robustness frontier of the final model.

Appendix D surveys related work.

**Notation and Problem Setup.** We consider a dataset $D$ where each labeled example $\{\mathbf{x}_i, y_i\}$ is associated with a discrete group label $g_i \in \mathscr{G} = \{1, \ldots, |\mathscr{G}|\}$. We denote $\mathscr{D} = P(y, \mathbf{x}, g)$ the joint distribution of the label, feature and groups, so that $D$ can be understood as a size-$n$ set of i.i.d. samples from $\mathscr{D}$ We denote the prevalence of each group as $\gamma_g = E_{(y, \mathbf{x}, g) \sim \mathscr{D}}(1_{G=g})$ and associate *dataset bias* with the imbalance in group distribution $P(G) = [\gamma_1, \ldots, \gamma_{|\mathscr{G}|}]$ (Rolf et al., 2021). In the applications we consider, there exists a subset of *underrepresented* groups $\mathscr{B} \subset \mathscr{G}$ which are not sufficiently represented in the population distribution $\mathscr{D}$ so that $\gamma_g \ll \frac{1}{|\mathscr{G}|}$ for $g \in \mathscr{B}$ (Sagawa et al., 2019; 2020). We denote $L(y, \hat{y})$ as a loss function from the Bregman divergence family, and $\mathscr{F}$ the hypothesis space of predictors $f : \mathscr{X} \mapsto \mathscr{Y}$. We require the model class $\mathscr{F}$ to be sufficiently expressive so it can model the Bayes-optimal predictor $\tilde{y}(\mathbf{x}) = \arg\min'_y E_{y \sim P(y|\mathbf{x})}(L(y, y'))$. We also assume $\mathscr{F}$ has a certain degree of smoothness, so that the model $f \in \mathscr{F}$ cannot arbitrarily overfit to the noisy labels in the training set.[1]

## 2 METHOD

In this section, we introduce *Introspective Self-play (*ISP*)*, a simple training approach to improve model quality in representation learning and uncertainty quantification under dataset bias. Briefly, ISP performs *introspective training* by adding a *underrepresention prediction* head to the model and training it to distinguish whether an example $(y_i, \mathbf{x}_i, g_i)$ is from the set of underrepresented groups $\mathscr{B}$ (Section 2.1). When the underrepresentation label $b_i = I(g_i \in \mathscr{B})$ is not available, ISP estimates it based on a cross-validation-based procedure we term *cross-validated self-play* (Section 2.2). As we will show, ISP carries a guarantee for the model's representation learning and uncertainty estimation quality under dataset bias (Proposition 1).

### 2.1 INTROSPECTIVE TRAINING

We consider models of the form $p(y|\mathbf{x}) = \sigma(f_y(\mathbf{x})) = \sigma(\beta_y^\top h(\mathbf{x}))$, where $h : \mathscr{X} \to \mathbb{R}^D$ is a $D$-dimensional embedding function, $\beta_y \in \mathbb{R}^D$ the output weights, and $\sigma(\cdot)$ the activation function. Given model $f_y = \beta_y^\top h$, *introspective training* adds a bias head $f_b = \beta_b^\top h$ to the model, so it becomes a multi-task architecture $f = (f_y, f_b)$ with shared embedding:

$$p(y|\mathbf{x}) = \sigma(f_y(\mathbf{x})), \ p(b|\mathbf{x}) = \sigma_{sigmoid}(f_b(\mathbf{x})); \text{ where } (f_y, f_b) = (\beta_y^\top h + b_y, \ \beta_b^\top h + b_b). \quad (3)$$

Given examples $D = \{\mathbf{x}_i, y_i, g_i\}_{i=1}^n$, we generate the underrepresentation labels as $b_i = I(g_i \in \mathscr{B})$ and train the model with the target and underrepresentation labels $(y_i, b_i)$ by minimizing a standard multi-task learning objective:

$$L((y_i, b_i), \mathbf{x}_i) = L(y_i, f_y(\mathbf{x}_i)) + L_b(b_i, f_b(\mathbf{x}_i)), \quad (4)$$

where $L$ is the standard loss function for the task, and $L_b$ is the cross-entropy loss. As a result, given training examples $\{\mathbf{x}_i\}_{i=1}^n$, *introspective training* not only trains the model to predict the outcome $y_i$, but also instructs it to recognize its potential bias $b_i$ by predicting whether $\mathbf{x}_i$ is from an underrepresented group.

Despite its simplicity, introspective training has a significant impact on the model's representation learning that is particularly important for quantifying uncertainty when dataset exhibits significant bias. Figure 2 illustrates this on a binary classification task under severe group imbalance (Sagawa et al., 2020), where we compare two dense ResNet ensemble models trained using the introspection objective v.s. the empirical risk minimization (ERM) objective (i.e., only use $L(y_i, f_y(\mathbf{x}_i))$ in Equation (4)), respectively.

Comparing figures 2a and 2e, we observe that the decision boundaries for the predicted label are very similar between introspective training and ERM. However, the predictive variance (obtained via a Gaussian process (GP) layer (Liu et al., 2022)) exhibits sizable differences. In particular, the variance estimates for introspective training are uniformly high outside of the two clouds of underrepresented groups in the data. However, for ERM, the model confidence is high along the decision boundary, even in the unseen regions without training data. This is due to the fact that when training with ERM, the representation collapses in the direction that is not correlated with training label (i.e., parallel

---

to decision boundary) and does not retrain any input information regarding the underrepresented groups in its representation (fig. 2g). However, with introspective training, the representations indeed are morphed to reflect the differences between the underrepresented examples and the majority group (as can be seen in figures fig. 2g vs fig. 2c), helping the model to better distinguish them in the representation space, and hence lead to improved uncertainty estimate in the neighborhood of underrepresented examples. Appendix E.1 contains further description.

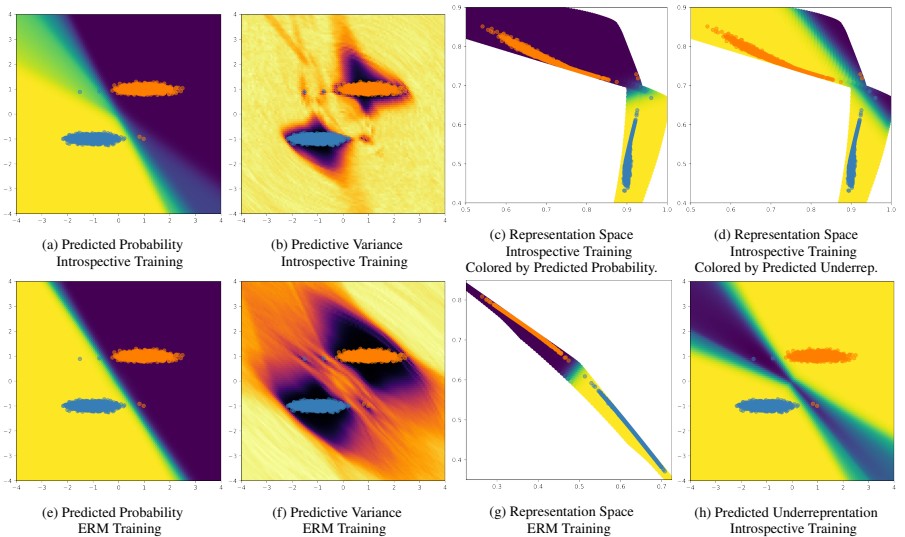

Figure 2: Prediction, uncertainty quantification, and representation learning behavior of introspective training v.s. ERM training in a binary classification task under severe group imbalance ($n = 5000$) (Sagawa et al., 2020). Here, blue and orange indicates the two classes, and each class contains a minority group (the tiny clusters on the diagonal with $n < 5$) and a majority group (the large clusters on the off-diagonal). **Column 1-2** depicts the models' predictive probability and predictive uncertain surface in the data space. **Column 3** depicts the models' decision surface in the last-layer representation space, colored by the predictive probability of the target label. **Column 4** depicts the introspective-trained model's predicted bias probability in the representation space (fig. 2d) and in the data space (fig. 2h), colored by the predictive probability of the underrepresentation. Appendix E.1 described further detail.

Formally, introspective training induces the below guarantee on the model's *bias-awareness* in its hidden representation and uncertainty estimates:

**Proposition 1** (Introspective Training induces Bias-awareness). *Denote $o_b(\mathbf{x}) = p(\mathbf{x}|b=1)/p(\mathbf{x}|b=0)$ the odds for $\mathbf{x}$ belongs to the underrepresented group $\mathcal{B}$. For a well-trained model $f = (f_y, f_b)$ that minimizes the introspective training objective (4), so that $p(b=1|\mathbf{x}) = \sigma(f_b(\mathbf{x}))$, we then have:*

*(I)* *(**Bias-aware Hidden Representation**) The hidden representation $h(\mathbf{x})$ is aware of the likelihood ratio of whether an example $\mathbf{x}$ belongs to the underrepresented group $b = I(g \in \mathcal{B})$, i.e. $p(\mathbf{x}|b=1)/p(\mathbf{x}|b=0)$, such that:*

$$\beta_b^\top h(\mathbf{x}) + b_b = \log o_b(\mathbf{x}) + \log \frac{p(b=1)}{p(b=0)}. \tag{5}$$

*(II)* *(**Bias-aware Embedding Distance**) For two examples $(\mathbf{x}_1, \mathbf{x}_2)$, the embedding distance $||h(\mathbf{x}_1) - h(\mathbf{x}_2)||_2$ is lower bounded by (up to a scaling constant) the odds ratio of whether $\mathbf{x}_1$ belongs to the underrepresented groups versus that for $\mathbf{x}_2$:*

$$||h(\mathbf{x}_1) - h(\mathbf{x}_2)||_2 \geq \frac{1}{||\beta_b||_2} \times \max\left(\log\frac{o_b(\mathbf{x}_1)}{o_b(\mathbf{x}_2)}, \log\frac{o_b(\mathbf{x}_2)}{o_b(\mathbf{x}_1)}\right), \tag{6}$$

*such that the distance between a pair of minority and majority examples $(\mathbf{x}_1, \mathbf{x}_2)$ is large due to the high values of the log odds ratio.*

The proof is in Appendix G. Part (I) provides a consistency guarantee for the hidden representation $h(\cdot)$'s ability in expressing the likelihood of whether an example $\mathbf{x}$ belongs to the underrepresented group $\mathcal{B}$, i.e., *bias awareness*. The form of (5) is similar to the representation learning guarantee in the noise contrastive learning literature, as it shares the same underlying principle of encouraging feature diversity and disentanglement via contrastive comparison between groups (Gutmann & Hyvärinen, 2010; Sugiyama et al., 2010; Hyvarinen & Morioka, 2016) . Part (II) is a corollary of (5) and provides a direct guarantee on the model's learned embedding distance. It states that under introspective training, the model cannot discard important input features that distinguishes

the minority-group examples from the majority group, even if they are not predictive of the target label. In this way, the model is guarded from collapsing the representation of majority and minority examples together (i.e., making $||h(\mathbf{x}_1) - h(\mathbf{x}_2)||_2$ excessively small for two examples $(\mathbf{x}_1, \mathbf{x}_2)$ from the majority and minority group, respectively), creating difficulty for identifying underrepresented groups in the feature space with uncertainty-based active learning. Empirically, we find the benefit of introspective training extends to other uncertainty-based active learning signals as well (e.g., margin and ensemble diversity, see Section 4.2). Appendix C contains further discussion.

## 2.2 Estimating unknown bias via Cross-validated Self-play

So far, we introduced *introspective training* in the setting where the group annotation $g_i$ is available on training data, so that the underrepresentation label $b_i = I(g_i \in \mathscr{B})$ can be directly computed. In this section, we consider how to estimate the underrepresentation label $b_i$ when it is absent, so that ISP can be applied to the setting where group annotations $g_i$ is too expensive to obtain. A popular practice in the literature is to estimate dataset bias as the predictive error of a single (biased) model. That is, given a trained model $f_D$, prior work (Clark et al., 2019; He et al., 2019; Nam et al., 2020; Sanh et al., 2020; Liu et al., 2021) estimates the underrepresentation label as the observed error $L(y_i, f_D(\mathbf{x}_i))$. To better understand this estimator for the generalization error of the underrepresented groups, we perform a noise-bias-variance decomposition (Domingos (2000)) of the model error $L(y, f_D)$, which reveals, in the expectation of the random draws of the dataset $D \sim \mathscr{D}$:

$$\underbrace{E_D[L(y, f_D(\mathbf{x}))]}_{error} = \underbrace{E_D\big[L(y, \tilde{y}(\mathbf{x}))\big]}_{noise} + \underbrace{L(\tilde{y}(\mathbf{x}), \bar{f}(\mathbf{x}))}_{bias} + \underbrace{E_D\big[L(\bar{f}(\mathbf{x}), f_D(\mathbf{x}))\big]}_{variance}, \qquad (7)$$

where $\tilde{y}(\mathbf{x}) = \arg\min_{y'} E_{y \sim P(y|\mathbf{x})}[L(y, y')]$ is the (Bayes) optimal predictor and $\bar{f}(\mathbf{x}) = \arg\min_f E_D[L(f, f_D(\mathbf{x}))]$ is the 'ensemble' predictor of the single models $\{f_D\}_{D \sim \mathscr{D}}$ trained from random data draws (see Appendix A.2 for a review). From (7), we see that for the purpose of estimating generalization error due to dataset bias, the naive estimator $\hat{b}_0 = L(y, f_D)$ based on single-model error suffers from two issues: (1) $\hat{b}_0$ conflates *noise* (typically arising from label noise or feature ambiguity) with the dataset bias signal we wish to capture, potentially leading to compromised quality in real datasets (Lahoti et al., 2020; Li et al., 2022). (2) As $\hat{b}_0$ is calculated from a single model, its estimate of the *variance* term (an important component of generalization error (Yang & Xu, 2020)) is often not stable. This is exacerbated when $\hat{b}_0$ is computed from the training error, since DNNs tend to severely underestimate the model variance on training data (Liu et al., 2021).[2]

This observation motivates us to propose ***cross-validated self-play***, a simple method to estimate a model's generalization gap. Briefly, given training data $D$ divided into $K$ splits, we train a bootstrap ensemble of $K$ models $\{f_k\}_{k=1}^K$ with ERM training, where each $f_k$ sees a fraction of the training data (see Appendix Fig. 5). As a result, for each $(\mathbf{x}_i, y_i)$, there exists a collection of in-sample predictions $\{f_{in,k'}(\mathbf{x}_i)\}_{k'=1}^{K_{in}}$ trained on data splits containing $(\mathbf{x}_i, y_i)$, and a collection of out-of-sample predictions $\{f_{out,k}(\mathbf{x}_i)\}_{k=1}^{K_{out}}$ trained on data splits not containing $(\mathbf{x}_i, y_i)$. Then, the ***self-play estimator*** of the model's generalization gap is [3]

$$\hat{b}_i = \underbrace{\mathbb{E}_k[L(y_i, f_{out,k}(\mathbf{x}_i))]}_{estimated\ error} - \underbrace{L(y_i, \bar{f}_{in}(\mathbf{x}_i))}_{estimated\ noise} = \mathbb{E}_k[L(\bar{f}_{in}(\mathbf{x}_i), f_{out,k}(\mathbf{x}_i))]. \qquad (8)$$

where $\bar{f}_{in}$ is the ensemble prediction based on in-sample predictors $f_{in,k'}$, the expectation $\mathbb{E}_k$ is taken with respect to the out-of-sample predictions, and we are estimating the Bayes optimal predictor $\tilde{y}$ using the in-domain prediction $\bar{f}_{in}$ (since the model class $\mathscr{F}$ is subject to suitable regularization, the $\bar{f}_{in}$'s do not arbitrarily overfit the noisy labels). Note that under the well-specified setting, $\bar{f}_{in}$ converges asymptotically to the Bayes optimal predictor $\tilde{y}$ as $n \to \infty$. In practice, we can ensure the validity of $L(y_i, \bar{f}_{in}(\mathbf{x}_i))$ as an estimator of noise by applying early stopping with a stability criterion based on out-of-sample predictions $\bar{f}_{out}(\mathbf{x}_i)$ (Li et al., 2020; Song et al., 2020). Compared to the standard alternatives in the literature (e.g., single-model error $L(y, f_D)$), the *self-play* estimator $\hat{b}_i$ has

---

[2]As an illustrative example, the generalization error of a ridge regression model under orthogonal design and group-specific noise is
$E_D(L(y, f_D(\mathbf{x}_g))) = \sigma_g^2 + \frac{(\lambda\theta_g)^2}{(n_g+\lambda)^2} + \frac{\sigma^2 n_g}{(n_g+\lambda)^2}$, where $\sigma_g$ is the noise level for group $g \in \mathscr{G}$, $n_g$ is the sample size for group $g \in \mathscr{G}$, and $\lambda$ is the ridge regularization parameter. See Appendix F for details.
[3]In this work, we use mean squared error $L(y, f) = \sqrt{(y - \sigma_{sigmoid}(f))^2}$ for the generalization gap computation, so that $\hat{b}_i \in [0, 1]$.

the appealing property of controlling *noise* (by using $\bar{f}_{in}$) while better estimating *variance* (by using expectations over $\bar{f}_{out,k}$), thereby constituting a more informative signal for the underrepresented groups under dataset bias, label noise and feature ambiguity. Appendix B.2 contains additional comments, and Appendix H develops a data-dependent bound for group detection performance.

**Method Summary: Introspective Self-play.** Combining the *self-play bias estimation* and *introspective training* together, we arrive at *Introspective Self-play* (ISP), a simple two-stage method that provably improves the representation quality and uncertainty estimates of a DNN for underrepresented population groups. Figure 3 illustrates the full ISP procedure. ISP first (optionally) estimates underrepresentation labels using *cross-validated self-play* if the group annotation is not available, and then conducts *introspective training* to train the model to recognize its own bias while learning to predict the target label. For the unlabelled data to be sampled, the resulting model generates (1) predictive probability $p(y|\mathbf{x})$, (2) uncertainty estimates $\hat{v}(\mathbf{x})$ and (3) predicted probability for underrepresentation $p(b|\mathbf{x}) = \sigma(f_b(\mathbf{x}))$, offering a rich collection of active learning signals for downstream applications.

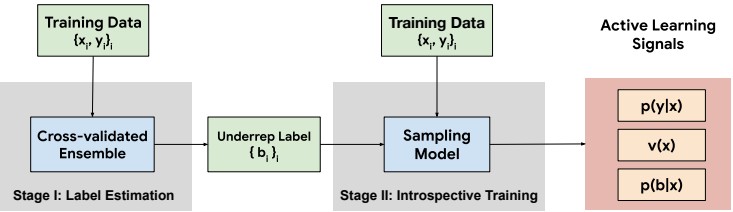

Figure 3: The two-stage *Introspective Self-play* (ISP) model: In the first stage, we first train a bootstrap ensemble of DNN's on cross-validation splits of the data (Figure 5). At the end of Stage I training, we can use the in-sample and out-of-sample predictions of the ensemble to estimate the underrepresentation label $b_i$ of each training data point $\mathbf{x}_i$ by computing the generalization gap (Equation (8)). Then in Stage II, we use the underrepresentation labels $b_i$ to train the actual sampling model via introspective training (Equation (4)), and use it to generate the active learning signals (e.g., predictive probability $p(y|x)$, predictive variance $v(x)$ and predicted underrepresentation $p(b|x)$) for the unlabelled set.

# 3 THEORETICAL ANALYSIS: IMPROVING ACCURACY-GROUP ROBUSTNESS FRONTIER BY OPTIMIZING TRAINING DATA DISTRIBUTION

Denote by $\Delta^K$ the $K$-simplex so that $\forall \alpha = [\alpha_1, \ldots, \alpha_K] \in \Delta^K$, $\sum_k \alpha_k = 1$ and $\forall k, \alpha_k > 0$. Let $D_{\alpha,n}$ denote a dataset of size $n$ sampled with group allocations $\alpha \in \Delta^{|\mathcal{G}|}$, i.e., $\alpha_g$ represents the fraction of the dataset sampled from group $g \in \mathcal{G}$. Our goal is to find the optimal allocation that minimizes a weighted combination of the population risk and the worst-case risk over all subgroups.

**Definition 1** (Risk and Fairness Risk). *Given $f \in \mathcal{F}$, let $R(f|\alpha,n) := \sum_{g \in \mathcal{G}} \gamma_g R(f|G=g)$ be the risk where $\gamma \in \Delta^{|\mathcal{G}|}$ represents the true subgroup proportions in the underlying data distribution, and $R_{fair}(f)$ be the fairness risk which is the worst-case risk among the groups $g \in \mathcal{G}$: $R_{fair}(f) := \max_g R(f|G=g)$.*

Let $\hat{R}$ and $\hat{R}_{fair}$ denote the empirical estimates of $R, R_{fair}$ estimated based on the finite dataset $D_{\alpha,n}$:

$$\hat{R}_{\alpha,n,g}(f) = \frac{\sum_{(x,y) \in D_{\alpha,n,g}} L(f(x),y)}{|D_{\alpha,n,g}|}, \hat{R}_{\alpha,n}(f) = \sum_g \gamma_g \hat{R}_{\alpha,n,g}(f), \hat{R}_{fair,\alpha,n}(f) = \max_g \hat{R}_{\alpha,n,g}(f)$$

where $D_{\alpha,n,g}$ is the subset of $D_{\alpha,n}$ with group label $g$. Let $\hat{f}$ denote the classifier obtained as per $\gamma$-weighted empirical risk minimization (ERM) $\hat{f}_{\alpha,n} = \arg\min_{f \in \mathcal{F}} \hat{R}_{\alpha,n}(f)$.

**Definition 2** (Accuracy-Fairness Frontier Risk). *For non-negative weighting coefficients $\omega \in [0,1]$, the accuracy-fairness frontier risk is defined as:*

$$F_\omega(\alpha,n) := \omega E[R(\hat{f}_{\alpha,n})] + (1-\omega)E[R_{fair}(\hat{f}_{\alpha,n})] \tag{9}$$

*where the expectation is taken wrt the randomness inherent in $\hat{f}$ as it is estimated based on a random dataset $D_{\alpha,n}$ drawn iid from the underlying data distribution.*

As we sweep over $\omega$ going from 0 to 1 and find the best classifier/allocation for each value, we would trace an accuracy fairness frontier as in Figure 1.

**Analyzing optimal allocation between subgroups.** We theoretically analyze the optimization problem $\alpha^* = \arg\min_{\alpha \in \Delta^{|\mathcal{G}|}} \left[ F_\omega(\alpha,n) \right]$. We build on the theoretical models for group-specific scaling

laws of generalization error introduced in previous work. (Chen et al., 2018; Rolf et al., 2021). We assume that the group-specific true risk decays with the size of the dataset at a rate of the form:

$$E[R(\hat{f}_{\alpha,n}|G=g)] = c_g(\alpha_g n)^{-p} + \tau n^{-q} + \delta$$

for some $p > 0, q > 0, c_g > 0, \tau, \delta > 0$. The first term represents the impact of the group representation, the second the aggregate impact of the dataset size, and the third term a constant offset (the irreducible risk for this group). In particular, $c_g$ represents the "difficulty" of learning group $g$, as the larger $c_g$ is, the higher the risk is for group $g$ for any given allocation $\alpha$.

**Theorem 1** (Optimal Group-size Allocation for Accuracy-Fairness Frontier Risk (Informal)). *The optimal allocation $\alpha^\star$ is of the form $\alpha_g^\star \propto (\gamma_g c_g \theta_g)^{\frac{1}{p+1}}$ where $\theta_g \geq \omega$ represents an up-sampling factor for group $g$. Let $g_1, g_2, \ldots$ be the subgroups sorted in ascending order according to the value $\gamma_g c_g^{-1/p}$, which represents the subgroup representation normalized by the difficulty of learning the subgroup. Then, it holds that there exists an integer $k > 0$ such that $\theta_{g_i} = \omega$ for $i = k, \ldots |\mathcal{G}|$, $\theta_{g_i} > \omega$ for $i = 1, \ldots, k$. Thus, the subgroups with low normalized representation are systematically up-sampled in the optimal allocation.*

Based on the above theorem, we can the set of underrepresented groups as $\mathcal{B} = \{g_1, g_2, \ldots g_k\}.$, validating the idea that underrepresented groups can be formally defined. While the result above is derived under a simplified theoretical model, it validates the intuition behind ISP: indeed, ISP attempts to infer whether datapoints belong to $\mathcal{B}$ and systematically up-samples them via active learning to increase the overall representation and achieve an allocation closer to the theoretically optimal allocation defined above. We present a full theorem statement and proof in Appendix I and a discussion in Appendix C.2

## 4 EXPERIMENTS

We first demonstrate that for each task, ISP meaningfully improves the tail-group sampling rate and the accuracy-group robustness performance of state-of-the-art AL methods (Section 4.1), and then conduct detailed ablation analysis to understand how the choice of different underrepresentation labels impacts (1) the final model's accuracy-group robustness performance when trained on data collected by different AL methods; and (2) the sampling performance of different active sampling signals (Section 4.2).

**Datasets.** We consider two challenging real-world datasets: Census Income (Le Quy et al., 2022) that contains 32,561 training records from the 1994 U.S. census survey. The task is to predict whether an individual's income is >50K, and the tail groups are female or non-white individuals with high income. We also consider Toxicity Detection (Borkan et al., 2019) that contains 405,130 online comments from the CivilComments platform. The goal is to predict whether a given comment is toxic, and the tail groups are demographic identities $\times$ label class (male, female, White, Black, LGBTQ, Muslim, Christian, other religion) $\times$ (toxic, non-toxic) following Koh et al. (2021).

| AL Training Method | Group identity label in train set? | Training Mechanism | Underrepresentation Label $b_i$ | Available Sampling Signal |
|---|---|---|---|---|
| (Random) | ✓ | - | Group Identity | Random |
| RWT (Idrissi et al., 2022) | ✓ | Reweighting | Group Identity | Margin / Diversity / Variance |
| DRO (Sagawa et al., 2019) | ✓ | Worst-group Loss | Group Identity | Margin / Diversity / Variance |
| ISP-Identity | ✓ | Introspection | Group Identity | Margin / Diversity / Variance / Predicted Underrep. |
| (ERM) | ✗ | - | Train Error | Margin / Diversity / Variance |
| JTT (Liu et al., 2021) | ✗ | Reweighting | Train Error | Margin / Diversity / Variance |
| ISP - Gap | ✗ | Introspection | Generalization Gap | Margin / Diversity / Variance / Predicted Underrep. |

Table 1: Training methods to be compared in the experiment study. Components proposed in this work are highlighted in red. The two baselines **(Random)** & **(ERM)** does not use underrepresentation label to train AL model, and only use it as reweighting signal for the reweighted training of the final model. For detailed definition of the sampling signals, see Appendix E.2.

**AL Baselines and Method Variations.** For all tasks, we use a 10-member DNN ensemble $f = \{f_k\}_{k=1}^{10}$ as the AL model, and replace their last layers with a random-feature GP layer (Liu et al., 2022) in order to compute posterior variance (see Appendix A.4). We compare the impact of different training methods in two settings depending on whether the group identity label will be annotated in the labelled set (they are *never* available in the unlabelled set). As shown in Table 1, when group label is available, we compare ISP-identity (i.e., ISP with group identity as training label $b_i = I(g_i \in \mathcal{B})$) to a group-specific reweighting (RWT) (Idrissi et al., 2022) and a group DRO (Sagawa et al., 2019) baselines (Idrissi et al., 2022) When the group label is not known, we consider **ISP-Gap** using the *self-play*-estimated generalization gap $\hat{b}_i = \mathbb{E}_k[L(\bar{f}_{in}(\mathbf{x}_i), f_{out,k}(\mathbf{x}_i))]$ as the representation label (i.e.,

Equation (8)), and compare it to an ensemble of Just Train Twice (JTT) which uses the ensemble training error $\hat{b}_i = \mathbb{E}_k[L(y_i, f_{in,k}(\mathbf{x}_i))]$ to determine the training set. We also compare to an ERM baseline which trains the AL models with a routine ERM objective, but uses error for the reweighted training of the final model. We consider other method combinations in the ablation study.

**Active Learning Protocol and Final Accuracy-Group Robustness Evaluation.** For active learning, we start with a randomly sampled initial dataset, and conduct active learning for 8 rounds until reaching roughly half of the training set (to ensure there's sufficient variation in the sampled data between methods). To evaluate the accuracy-group robustness performance of the final model, given a dataset collected by an AL method, we train a final model using the standard re-weighting objective $\sum_{(x,y)\notin\hat{\mathscr{B}}} L_{ce}(y, f(\mathbf{x})) + \lambda \sum_{(x,y)\in\hat{\mathscr{B}}} L_{ce}(y, f(\mathbf{x}))$ where $\hat{\mathscr{B}}$ is the set of underrepresented examples identified by the underrepresentation label, i.e., $(x_i, y_i) \in \hat{\mathscr{B}}$ if $\hat{b}_i > t$. We vary the thresholds $t$ and the up-weight coefficient $\lambda$ over a 2D grid to get a collection of model accuracy-group robustness performances (i.e., accuracy v.s. worst-group accuracy), and use them to identify the Pareto frontier defined by this combination of data and reweighting signal. Appendix E.2 describes further detail.

| AL Training Method | Group identity label in train set? | Census Income | | | Toxicity Detection | | |
|---|---|---|---|---|---|---|---|
| | | Tail Sampling Rate | Combined Acc. | Worst-group Acc. | Tail Sampling Rate | Combined Acc. | Worst-group Acc. |
| (Random) | ✓ | 0.475 | 0.746 | 0.659 | 0.556 | 0.708 | 0.490 |
| RWT | ✓ | 0.797 | 0.772 | 0.761 | 0.857 | 0.709 | 0.482 |
| DRO | ✓ | 0.755 | 0.759 | 0.729 | 0.841 | 0.710 | 0.506 |
| ISP-Identity (Ours) | ✓ | **0.907** | **0.785** | **0.774** | **0.905** | **0.719** | **0.506** |
| ERM | × | 0.791 | 0.736 | 0.658 | 0.852 | 0.735 | 0.539 |
| JTT | × | **0.839** | 0.752 | 0.695 | 0.866 | 0.747 | 0.571 |
| ISP-Gap (Ours) | × | **0.839** | **0.770** | **0.753** | **0.867** | **0.759** | **0.597** |

Table 2: The tail-group sampling rate and final-model accuracy v.s. group robustness performances under different AL model training methods. Here we show the best active learning signal for each task (i.e., variance for Census Income, and margin for toxicity detection). **Tail Sampling Rate**: The ratio between num. of sampled tail group examples (in final round) v.s. the total num. of tail group in population. **Combined Acc**: The combined accuracy-robustness score defined as (acc + worst-group acc)/2. It is proportional to the perimeter of the rectangle defined by a point on the accuracy-group robustness curve.

## 4.1 MAIN RESULTS

Table 2 shows sampling performance and the final-model group robustness-accuracy performance of each AL model training method (described in Table 1), and Figure 4 visualizes the full accuracy-fairness frontier of the final models (trained on the data and re-weighting signals provided by that method). Our main conclusions are: **(1) Effectiveness of ISP training**: Compared to non-ISP baselines, we find ISP consistently improves a AL model's active learning (measured by tail-group sampling rate) and accuracy-group robustness performance (measured by combined

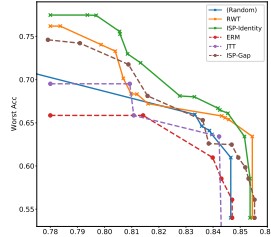

Figure 4: Accuracy-Group Robustness Frontier for Census Income.

accuracy, which is defined as (accuracy + worst-group accuracy)/2). This advantage is seen in both settings where the group label is available or unavailable. In particular, in Figure 4, the final model from **ISP-Gap** (pink dashed line, trained on actively sampled data and using estimated underrepresentation label for final-model re-weighted training) almost dominates **Random** (blue solid line, trained on randomly sampled data and using true group label for final model training) despite not having access to true group label in the final reweighted training, highlighting the importance of the data distribution in the model's accuracy-group robustness performance (i.e., Equation (2)). **(2) Label Quality Matters**: Comparing the variants of ISP (Identity v.s. Gap) in Table 2, we see a clear impact of the quality of introspection signal to the performance of the AL model. For example, for active learning performance, we see that the sampling rate ISP-Identity is significantly better than ISP-Error. However, for toxicity detection where the group label suffers an under-coverage issue (i.e., the group definition excludes potentially identity-mention comments where raters disagree, see **Data** section of Appendix E.2), we see that ISP-Error in fact strongly outperforms ISP-Identity in accuracy-group robustness performance. This validates the observation from previous literature on the failure mode of bias-mitigation methods when the available group annotation does not cover all sources of dataset bias, and speaks to the importance of high quality estimation methods that can detect underpresentation in the presence of unknown sources of bias (Zhou et al., 2021).

## 4.2 ABLATION ANALYSIS

In the main results above, we have (1) used the same underrepresentation label for both the AL-model introspective training and the final-model reweighted training, and (2) focused on the most effective active sampling signal under each task. In this section, we conduct ablations by decoupling the signal combinations along these two axes.

**Impact of Data Distribution and Reweighting Signal to Accuracy-Group Robustness Frontier**
First, we investigate the joint impact of data distribution and reweighting signal on the final models' accuracy-group robustness performance. We train the final model under data collected by different AL policy (Random v.s Margin v.s. Group Identity, etc), and perform reweighted training using different underrepresentation labels (Error v.s. Gap v.s. Group Identity) and compare to an ERM baseline without reweighted training (Section 4.2). As shown, holding the choice of reweighted signal constant and compare across data distributions (i.e., comparing across columns within each row), we observe that the data distribution in general has a non-trivial impact on the final model's accuracy-group robustness performance. Specifically, under appropriate sampling signal, data collected by ISP-Gap (which has no access to true group identity label) can lead to model performance that is competitive with data collected by ISP-Identity (e.g., the third v.s. fourth columns). Comparing across reweighting signals within each dataset (i.e., compare across rows within each column), we see that all underrepresentation labels brings a meaningful improvement over the ERM baseline, with Group Identity bringing the most significant improvement when it is of high quality (i.e., Census Income), and Gap bringing the most improvement when group annotation is imperfect (i.e., Toxicity Detection).

| Final Model Reweighting Signal | AL Method, Census Income | | | | AL Method, Toxicity Detection | | | |
|---|---|---|---|---|---|---|---|---|
| | Random | Margin | Variance | Group Identity | Random | Diversity | Margin | Group Identity |
| (ERM) | 0.692 | 0.669 | 0.719 | 0.720 | 0.698 | 0.699 | 0.702 | 0.703 |
| Error | 0.706 | 0.683 | 0.750 | 0.743 | 0.758 | 0.761 | 0.744 | 0.752 |
| Gap | 0.692 | 0.694 | 0.770 | 0.777 | **0.776** | **0.776** | **0.758** | **0.810** |
| Group Identity | **0.746** | **0.756** | **0.778** | **0.785** | 0.711 | 0.701 | 0.705 | 0.713 |

Table 3: Impact to final-model group robustness-accuracy performance (measured by combined accuracy = acc + worst-group acc)/2) of the choice of reweighting signal (rows), across dataset collected by different active learning methods (columns). **Random**: data collected via random sampling. **Margin/Variance/Diversity**: data collected using introspective-trained AL model (with Gap as underrepresentation label) using the said sampling signal. **Group Identity**: data collected by introspective-trained AL model with group identity as introspection signal, using the best sampling signal for the task (Variance for census income and Margin for toxicity detection).

**Impact of Underrepresentation Label on Different Sampling Signals.** Finally, we evaluate the choice of introspection signal on the sampling performance of a introspective-trained AL-model, under different types of sampling signals (Table 4). As this evaluation is computationally expensive (requiring multiple active learning experiments for all underrepresentation label v.s. sampling signal combinations), here we focus on the Census Income task. As shown, we observe the introspective training brings a consistent performance boost across different types of sampling signals (esp. when using Group Identity), highlighting the appeal of introspective training as a "plug-in" method that meaningfully boost the performance of a wide range of active learning methods. Interestingly, we also observe the "Predicted Underrep." (i.e., the underrepresentation prediction in $p(b|\mathbf{x})$ in Figure 3) is exceptionally effective when the group identity is available (tail sampling rate $> 0.95$) but underperforms classic active learning signals otherwise, cautioning the proper use of $p(b|\mathbf{x})$ as a sampling signal depending on the availability of group labels.

| Underrep. Label | AL Method, Census Income | | | |
|---|---|---|---|---|
| | Margin | Diversity | Variance | Predicted Underrep. |
| Error | 0.780 | 0.324 | 0.771 | 0.671 |
| Gap | 0.803 | 0.276 | 0.839 | 0.708 |
| Group Identity | 0.873 | 0.330 | 0.907 | 0.967 |

Table 4: Impact to AL performance (measured by tail sampling rate) of the choice of introspection signal (rows) across different active learning methods (columns).

## 5 CONCLUSION

In this work, we introduced *Introspective Self-play* (ISP), a novel training approach to improve a DNN's representation learning and uncertainty quantification quality under dataset bias. ISP uses a multi-task *introspective training* approach to encourage DNNs to learn diverse and bias-aware features for the underrepresented groups. When underrepresented group identities are not available, ISP bootstraps them using a novel *cross-validated self-play* procedure that disentangles dataset bias from irreducible noise while also generating a more stable estimate of variance. Theoretical analysis reveals that the optimal per-group up-sampling factors are in fact determined by an interplay of the original group rates and the group-specific scaling factors. Models trained on data acquired by ISP generally surpass recent competitive baselines such as RWT and JTT on the accuracy-group robustness frontier.

**Future Directions.** Overall, our results are a concrete step to a recent but critical effort in the community to build a more holistic perspective of model performance, addressing key challenges of robustness and equity. Future work could more thoroughly investigate the relation of the noise, bias, and variance components of generalization error to underrepresentation examples, as well as the effectiveness of introspective training under different settings including training epochs, model regularization, batch size. For example, specialized training objectives such as generalized cross entropy (GCE) (Zhang & Sabuncu, 2018) or focal loss (FL) (Lin et al., 2017) may improve the statistical power of different components of generalization error in detecting the underrepresented groups. On the other hand, batch size may impact the quality of the learned representation under introspective training, in a manner analogous to that of the contrastive training (Chen et al., 2020a). Our framework could also be extended to other settings such as semi-supervised learning, or incorporate other kinds of introspection signals, such as those from the interpetability or differential privacy literature.

**Ethical Statements.** This work proposes novel approach to encourage DNN models to learn diverse, *bias-aware* features in model representation, for the purpose of improving DNN model's uncertainty quantification ability under dataset bias. Our method encourages DNNs to better identify underrepresented data subgroups during data collection, and eventually achieve a more balanced performance between model accuracy and group robustness by training on a more well-balanced dataset. We evaluated the method on two already publicly available dataset and uses existing metrics in the literature. No new data is collected as part of the current study.

The technique we developed in this work is simple and general-purpose, with potentially broad appeal to various downstream applications (e.g., recommendations, NLP, etc). However, two limitations highlighted by our work is that (1) when the group annotation information is imperfect, building a bias-mitigation procedure around such annotation may lead to suboptimal performance, and (2) in the presence of noisy labels, a noisy estimate of under-representation may also compromise the performance of the procedure. Therefore, practitioner should take caution in rigorously evaluate the effectiveness of the procedure in their application, taking effort to carefully evaluate the estimation result of underrepresentation labels to ensure proper application of the technique without incurring unexpected consequences.

ACKNOWLEDGMENTS

The authors would like to sincerely thank Ian Kivlichan at Google Jigsaw, Clara Huiyi Hu, Jie Ren, Yuyan Wang, Tania Bedrax-Weiss at Google Research, and Martin Strobel at National University of Singapore for the insightful comments and helpful discussions.

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

## A  ADDITIONAL BACKGROUND

### A.1  RECAP: NOTATION AND PROBLEM SETUP.

**Dataset with subgroups:** We consider a dataset $D$ where each example $\{\mathbf{x}_i, y_i\}$ ($\mathbf{x}_i \in \mathcal{X}$ denotes the features and $y_i \in \mathcal{Y}$ the label) is associated with a discrete group label $g_i \in \mathcal{G} = \{1, \ldots, |\mathcal{G}|\}$.

**Joint data distribution:** We denote $\mathcal{D} = P(y, \mathbf{x}, g)$ as the joint distribution of the label, feature and groups, so that $D$ above can be understood as a size-$n$ set of i.i.d. samples from $\mathcal{D}$. Notice that this formulation implies a flexible noise model $P(y|\mathbf{x}, g)$ that depends on $(\mathbf{x}, g)$. It also implies a flexible group-specific distribution $P(y, \mathbf{x}|g)$, where the joint distribution of $(y, \mathbf{x})$ varies by group. Note however that we do assume that the group label does not have additional predictive power beyond appropriate representation of the features, i.e., there exists a representation function $h^*$ such that for $\mathbf{z} = h^*(\mathbf{x})$, we have $P(y|\mathbf{z}, g) = P(y|\mathbf{z})$, i.e., the semantic, label-relevant features are invariant across subgroups (Arjovsky et al., 2019; Creager et al., 2021; Shui et al., 2022). Note however that we do assume that the group label does not have additional predictive power beyond appropriate representation of the features, i.e., there exists a representation function $h^*$ such that for $\mathbf{z} = h^*(\mathbf{x})$, we have $P(y|\mathbf{z}, g) = P(y|\mathbf{z})$, i.e., the semantic, label-relevant features are invariant across subgroups (Arjovsky et al., 2019; Creager et al., 2021; Shui et al., 2022)

**Subgroup prevalence:** We denote the prevalence of each group as $\gamma_g = E_{(y, \mathbf{x}, g) \sim \mathcal{D}}(1_{G=g})$. As a result, the notion of *dataset bias* is reflected as the imbalance in group distribution $P(G) = [\gamma_1, \ldots, \gamma_{|\mathcal{G}|}]$ (Rolf et al., 2021). In the applications we consider, it is often feasible to identify a subset of *underrepresented* groups $\mathcal{B} \subset \mathcal{G}$ which are not sufficiently represented in the population distribution $\mathcal{D}$ and have $\gamma_g \ll \frac{1}{|\mathcal{G}|}$ (Sagawa et al., 2019; 2020). To this end, we also specify $\mathcal{D}^* = P(y, \mathbf{x}|g)P^*(g)$ an optimal distribution, where $P^*(g)$ is an ideal group distribution (i.e., uniform such that $P^*(g) = \gamma_g^* = \frac{1}{|\mathcal{G}|}$) so that all groups have sufficient representation in the data.

**Loss function:** We assume a loss function $L(y, \hat{y})$, that denotes the loss incurred when the predicted label is $\hat{y}$ while the actual label is $y$.

**Hypothesis space:** We consider learning a predictor from a hypothesis space $\mathcal{F}$ of functions $f : \mathcal{X} \mapsto \mathcal{Y}$. We assume that the hypothesis space is well-specified, i.e., that it contains the Bayes-optimal predictor $\tilde{y} : \mathcal{X} \to \mathcal{Y}$:

$$\tilde{y}(\mathbf{x}) = \arg\min_{y' \in \mathcal{Y}} E_{y \sim P(y|\mathbf{x})}(L(y, y')).$$

We require the model class $\mathcal{F}$ to come with certain degree of smoothness, so that the model $f \in \mathcal{F}$ cannot arbitrarily overfit to the noisy labels during the course of training. In the case of over-parameterized models, this usually implies $\mathcal{F}$ is subject to certain regularization that is appropriate for the model class (e.g., early stopping for SGD-trained neural networks) (Li et al., 2020).

### A.2  DISENTANGLING MODEL ERROR UNDER NOISE AND BIAS

Given a dataset $D \sim \mathcal{D}$ and a loss function $L$, we consider learning the prediction function $f_D = \arg\min_{f \in \mathcal{F}} L(f, y|D)$, where $L(y, f|D) = \sum_{\{\mathbf{x}_i, y_i\} \in D} L(y_i, f(\mathbf{x}_i))$. Following the previous work (Pfau, 2013), we denote the *ensemble predictor* $\bar{f} = \arg\min_{f \in \mathcal{F}} E_{D \sim \mathcal{D}}(L(f_D, f))$ over ensemble members $f_D$'s, where each $f_D$ is trained on a random draw of training dataset $D \sim \mathcal{D}$, and $\tilde{y}(\mathbf{x}) = \arg\min_{y' \in \mathcal{Y}} E_{y \sim P(y|\mathbf{x})}(L(y, y'))$ the (Bayes) optimal predictor. For test example $\{y_i, \mathbf{x}_i\}$, we can decompose the predictive error of a trained model $f_D(\mathbf{x})$ using a generalized bias-variance decomposition for Bregman divergence:

**Proposition A.1** (Noise-Bias-Variance Decomposition under Bregman divergence (Domingos, 2000; Pfau, 2013)). *Given a loss function of the Bregman divergence family, for a test example $\{y, \mathbf{x}\}$ the expected prediction loss $L(y, f_D(\mathbf{x}))$ of an empirical predictor $f_D$ can be decomposed as:*

$$E_D\big[L(y, f_D(\mathbf{x}))\big] = \underbrace{E_D\big[L(y, \tilde{y}(\mathbf{x}))\big]}_{\textit{Noise}} + \underbrace{L(\tilde{y}(\mathbf{x}), \bar{f}(\mathbf{x}))}_{\textit{Bias}} + \underbrace{E_D\big[L(\bar{f}(\mathbf{x}), f_D(\mathbf{x}))\big]}_{\textit{Uncertainty}} \quad (10)$$

Given a fixed data distribution $\mathcal{D}$, the first term $E_D\big[L(y, f^*(\mathbf{x}))\big]$ quantifies the *irreducible noise* that is due to the stochasticity in the noisy observation $y$. The third term $E_D\big[L(\bar{f}(\mathbf{x}), f_D(\mathbf{x}))\big]$ quantifies the *variance* in the prediction, which can be due to variations in the finite-size data $D$, the stochasticity in the randomized learning algorithm $\mathcal{F} \times D \to f_D$, or the randomness in the initialization of an overparameterized model (Adlam & Pennington, 2020). Finally, the middle term $L(\tilde{y}(\mathbf{x}), \bar{f}(\mathbf{x}))$

quantifies the *bias* between $\tilde{y}(\mathbf{x})$ (i.e., the "true label") and the ensemble predictor $\bar{f}$ learned from the empirical data $D \sim \mathscr{D}$. It is inherent to the specification of the model class and cannot be eliminated by ensembling, e.g., it can be caused by model misspecification, missing features, or regularization. To make the idea concrete, consider a simple example where we fit a ridge regression model $f(\mathbf{x}_i) = \beta^\top \mathbf{x}_i$ to the Gaussian observation data $y_i = \theta^\top \mathbf{x}_i + \varepsilon, \varepsilon \sim N(0, \sigma^2)$ under an imbalanced experiment design, where we have $|\mathscr{G}|$ treatment groups and $n_g$ observations in each group. Here, $\mathbf{x}_i = [1_{g_i=1}, ..., 1_{g_i=|\mathscr{G}|}]$ is a $|\mathscr{G}| \times 1$ one-hot indicator of the membership of $g_i$ for each group in $\mathscr{G}$, and $\theta = [\theta_1, ..., \theta_{|\mathscr{G}|}]$ is the true effect for each group. Then, under ridge regression, the noise-bias-variance decomposition for group $g$ is $E_D(L(y, f_D)) = \sigma^2 + \frac{(\lambda \theta_g)^2}{(n_g+\lambda)^2} + \frac{\sigma^2 n_g}{(n_g+\lambda)^2}$, where the regularization parameter $\lambda$ modulates a trade-off between the *bias* and *variance* terms.

## A.3 FURTHER DECOMPOSITION

**Further Uncertainty Decomposition for Probabilistic Models** As an aside, when the predictive model $f_D$ is probabilistic (e.g., the model generates a posterior predictive distribution $P(f|D)$ rather than a point estimate $f$), the *variance* in Equation (10) is further decomposed as:

$$E_D\big[L(\bar{f}(\mathbf{x}), f_D(\mathbf{x}))\big] = \underbrace{E_D\big[L(\bar{f}(\mathbf{x}), \mu_D(\mathbf{x}))\big]}_{\text{Ensemble Diversity}} + \underbrace{E_D E_{f \sim P(f|D)}\big[L(\mu_D(\mathbf{x}), f(\mathbf{x}))\big]}_{\text{Posterior Variance}} \qquad (11)$$

where $\mu_D(\mathbf{x}) = E_{f \sim P(f|D)}[f(\mathbf{x})]$ is the posterior mean, and $v_D(\mathbf{x}) = E_{f \sim P(f|D)}\big[L(\mu_D(\mathbf{x}), f(\mathbf{x}))\big]$ is the posterior variance of each ensemble member. As shown, comparing to an ensemble of deterministic models, the ensemble of probabilistic models provides additional flexibility in quantifying model uncertainty via the extra term of expected posterior variance.

**Further Bias Decomposition for Minority Groups** For the examples $\mathbf{x}$ coming from the under-represented groups with $\gamma_g \ll \frac{1}{|\mathscr{G}|}$, the bias term can be further decomposed into:

$$L(\tilde{y}(\mathbf{x}), \bar{f}(\mathbf{x})) = \underbrace{L(\tilde{y}(\mathbf{x}), \bar{f}^*(\mathbf{x}))}_{\text{Bias, Model}} + \underbrace{\mathscr{E}(\bar{f}^*(\mathbf{x}), \bar{f}(\mathbf{x}))}_{\text{Excess Bias, Data}}, \qquad (12)$$

where $\bar{f}^* = \arg\min_{f \in \mathscr{F}} E_{D^* \sim \mathscr{D}^*}(L(f_{D^*}, f))$ is the optimal ensemble predictor based on size-$n$ datasets $D^*$ sampled from the optimal distribution $\mathscr{D}^*$ where all groups have equal representation. Here, $L(\tilde{y}(\mathbf{x}), \bar{f}^*(\mathbf{x}))$ is the bias inherent to the model class and cannot be eliminated by ensembling. It can be caused by model misspecification, missing features, or regularization. On the other hand, $\mathscr{E}(\bar{f}^*(\mathbf{x}), \bar{f}(\mathbf{x})) = L(\tilde{y}(\mathbf{x}), \bar{f}(\mathbf{x})) - L(\tilde{y}(\mathbf{x}), \bar{f}^*(\mathbf{x}))$ indicates the "excess bias" for the underrepresented groups caused by the imbalance in the group distribution $P(G)$ in the data-generation distribution $\mathscr{D}$.

To make the idea concrete, consider the ridge regression example from the previous section, where the noise-bias-variance decomposition for group $g$ is $E_D(L(y, f_D)) = \sigma^2 + \frac{(\lambda \theta_g)^2}{(n_g+\lambda)^2} + \frac{\sigma^2 n_g}{(n_g+\lambda)^2}$, with the regularization parameter $\lambda$ modulating a trade-off between the *bias* and *variance* terms (Appendix F). Consequently, for an underrepresented group with small size $\gamma_g \ll \frac{1}{|\mathscr{G}|}$, its predictive bias $\frac{(\lambda \theta_g)^{*2}}{(n_g+\lambda)^2}$ is exacerbated due to lacking sufficient statistical information to counter the regularization bias, incuring an excessive bias of $\mathscr{E}(\bar{f}^*(\mathbf{x}), \bar{f}(\mathbf{x})) \approx \frac{\lambda \theta_g}{n \gamma_g^* \gamma_g}(\gamma_g^* - \gamma_g)$ when compared to an optimal ensemble predictor $\bar{f}^*$ trained from a perfectly balanced size-$n$ datasets with $\gamma_g^* = 1/|\mathscr{G}|$.

## A.4 MODERN UNCERTAINTY ESTIMATION TECHNIQUES IN DEEP LEARNING

For a deep classifier $p(\mathbf{x}) = \sigma(f(\mathbf{x}))$ with logit function $f(\mathbf{x}) = \beta^\top h(\mathbf{x})$ and $h(\mathbf{x}) \in \mathbb{R}^M$ the last-layer hidden embeddings, the modern deep uncertainty methods quantifies model uncertainty by enabling it to generate random samples from a predictive distribution. That is, for a model trained on data $D = \{(\mathbf{x}_i, y_i)\}_{i=1}^n$, given a test data point $\mathbf{x}_{test}$, the model can return a size-$K$ sample:

$$\{f_k(\mathbf{x}_{test})\}_{k=1}^K \sim P(f|\mathbf{x}_{test}, D).$$

For example, in Monte Carlo Dropout (Gal & Ghahramani, 2016), the samples is generated by perturbing the dropout mask in the learned predictive function $f(\cdot) = \beta^\top h(\cdot)$'s embedding function

$h(\cdot)$, while in Deep Ensemble (Lakshminarayanan et al., 2017), the sample comes directly from the multiple parallel-trained ensemble members. Finally, in a neural Gaussian process model (Wilson et al., 2016; Liu et al., 2022; van Amersfoort et al., 2021), the samples are generated from a Gaussian process model using the hidden embedding function $h(\mathbf{x})$ as the input. For example, for classification problems, the predictive variance of the Gaussian process model $v(\mathbf{x}_{test}) = Var(f|\mathbf{x}_{test}, D)$ can be expressed as (Williams & Rasmussen (2006), Chapter 3):

$$v(\mathbf{x}_{test}) = \mathbf{k}(\mathbf{x}_{test})_{1 \times n}^{\top} \mathbf{V}_{n \times n} \mathbf{k}(\mathbf{x}_{test})_{n \times 1};$$

where $\mathbf{V}_{n \times n}$ is a fixed matrix computed from training data, and $\mathbf{k}(\mathbf{x}_{test}) = [k(\mathbf{x}_{test}, \mathbf{x}_1), \ldots, k(\mathbf{x}_{test}, \mathbf{x}_n)]$ is a vector of kernel distances between $\mathbf{x}_{test}$ and the training examples $\{\mathbf{x}_i\}_{i=1}^n$. The kernel function $k$ is commonly defined to be a monotonic function of the hidden embedding distance, e.g., $k(\mathbf{x}_{test}, \mathbf{x}_i) = exp(-||h(\mathbf{x}_{test}) - h(\mathbf{x}_i)||_2^2)$ for the RBF kernel. As a result, the predictive uncertainty for a data points $\mathbf{x}_i$ is determined by the distance between $\mathbf{x}_{test}$ from the training data $\{\mathbf{x}_i\}_{i=1}^n$. Consequently, a DNN model's quality in representation learning has non-trivial impact on its uncertainty performance. Although first mentioned in the context of neural Gaussian process, this connection between the quality of representation learning and the quality of uncertainty quantification also holds for state-of-the-art techniques such as Deep Ensemble, as model averaging cannot eliminate the systematic errors in representation learning and consequently the issue in uncertainty quantification (for example, see Figure 6 and the corresponding ensemble uncertainty surface Figure 2f).

**Neural Gaussian Process Ensemble** In this work, to comprehensively investigate the effect of different uncertainty techniques, we should to use a Deep Ensemble of neural Gaussian process as our canonical model. That is, we parallel train $K$ neural Gaussian process models $\{f_k\}_{k=1}^K$. Then, given a test data point $\mathbf{x}_{test}$, each ensemble member will return a predictive distribution with means $\{\mu_k(\mathbf{x})\}_{k=1}^K$ and variances $\{v_k(\mathbf{x})\}_{k=1}^K$. Then, we can generate model prediction as $\mathbb{E}_k[\mu_k(\mathbf{x})]$, and quantify uncertainty in one of the two ways:

$$\text{Ensemble Diversity}: \quad Var_k(\mu_k(\mathbf{x}));$$
$$\text{Posterior Variance}: \quad \mathbb{E}_k(v_k(\mathbf{x})),$$

where $Var_k$, $\mathbb{E}_k$ are empirical means and variances over the ensemble members. As shown, they correspond to the two components of the total model variance under squared error introduced in A.3. We investigate the effectiveness of these two uncertainty signals in the experiments.

## A.5 CONNECTION BETWEEN GROUP ROBUSTNESS AND FAIRNESS

The notion of group robustness (e.g., worst-group accuracy) we considered in this work corresponds to the notion of *minimax fairness* or *Rawlsian max-min fairness* (Lahoti et al., 2020; Martinez et al., 2020; 2021; Diana et al., 2021). The philosophical foundation of this notion has been well-established (Rawls, 2001; 2004). There have been notable works developed for both analyzing accuracy-fairness tradeoff under this notion (e.g., Martinez et al. (2020; 2021)), and for improving fairness performance without explicit annotation (i.e., ARL, (Lahoti et al., 2020)). It is also identified as the original objective of the well-known *Invariant Risk Minimization* (IRM) work for invariant learning (Arjovsky et al. (2019), Section 2).

To this end, under a sampling model with well-calibrated uncertainty, the active learning procedure as discussed in introduction is expected to carry benefit for model fairness as well. Specifically, in the context of fairness-aware learning, this means a sampling model with calibrated uncertainty preferentially samples the under-represented protected group, leading to a sampled training set with more balanced representation among population subgroups, and consequently reducing the between-group generalization gap in the trained model and promoting its fairness properties.

# B  METHOD SUMMARY

## B.1  ALGORITHM

---

**Algorithm 1** *Introspective Self-play* (ISP)

---

**Inputs:** Training data $D_{train} = \{y_i, \mathbf{x}_i\}_{i=1}^n$; (Optional) Group annotation $G_{train} = \{g_i\}_{i=1}^n$;
        Unlabelled data $D_{pool} = \{\mathbf{x}_j\}_{j=1}^{n'}$.

**Output:** Predicted probability $\{p(y|\mathbf{x}_j)\}_{j=1}^{n'}$; Bias probability $\{p(b|\mathbf{x}_j)\}_{j=1}^{n'}$; Predictive variance $\{v(\mathbf{x}_j)\}_{j=1}^{n'}$.

▷ Stage I: Label Generation
**if** $G_{train} \neq \emptyset$ **then**
    $B_{train} = \{b_i = I(g_i \in \mathscr{B})\};$                     ▷ Make underrepresentation label using group annotation $g_i$.
**else**
    $\hat{B}_{train} = SelfPlayBiasEstimation(D_{train}).$    ▷ Estimate underrepresentation label using Algorithm 2

▷ Stage II: Introspective Training
Train $\hat{f}$ on $D_{train}$ with multi-task introspective objective $L((y_i, b_i), \mathbf{x}_i)$.          ▷ Equation (4)
Evaluate $\hat{f}$ on $\mathbf{x}_j \in D_{pool}$ to generate sampling signals $\{p(y|\mathbf{x}_j), p(b|\mathbf{x}_j), v(\mathbf{x}_j)\}_{j=1}^{n'}$.    ▷ Equation (3)

---

**Algorithm 2** Underrepresentation Label Estimation via *Cross-validated Self-play*

---

**Inputs:** Training data $D_{train} = \{y_i, \mathbf{x}_i\}_{i=1}^n$.
**Output:** Estimate underrepresentation labels $\hat{B}_{train}$.

Train $K$-fold cross-validated ensemble $\{f_k\}_{k=1}^K$ with $D_{train}$.
Compute in-sample and out-of-sample ensemble predictions $\{f_{in,k}(\mathbf{x}_i)\}_{k=1}^{K_{in}}, \{f_{out,k}(\mathbf{x}_i)\}_{k=1}^{K_{out}}$ for all $\mathbf{x}_i \in D_{train}$.
Estimate underrepresentation labels as $\hat{B}_{train} = \{b_i = \mathbb{E}_k[L(\bar{f}_{in}(\mathbf{x}_i), f_{out,k}(\mathbf{x}_i))]\}_{i=1}^n$.    ▷ (Equation (8))

---

## B.2  ESTIMATING GENERALIZATION GAP USING CROSS-VALIDATED ENSEMBLE

**Practical Comments.** Note that due to its cross validation nature, the *self-play* bias estimator $\hat{b}_i$ estimates the generalization error of a weaker model (i.e., trained on a smaller data size $n_{cv} < n$). This is in fact consistent with the practice in the previous debasing literature, where the main model is trained on the error signals from weaker and more biased models (Clark et al., 2019; He et al., 2019; Nam et al., 2020).

Further, in the context of SGD-trained neural networks, it is important to properly estimate the $\bar{f}_{in}(\mathbf{x}_i)$ so it does not overfit to the training label, via early stopping (Li et al., 2020; Liu et al., 2020). This is easy to do in the context of cross validation: during training, we collect the estimated bias $\hat{b}_{i,t}$ across the training epochs $t = 1, \ldots, T$, and select the optimal stopping point $t$ as the first time the out-of-sample error $\mathbb{E}[L(y_i, f_{out,k}(\mathbf{x}_i))]$ stablizes. In practice, we specify the early-stopping criteria as when the running average (within a window $T' = 5$) of the cross validation error first stablizes below a threshold $\varepsilon$. This is to prevent the situation where the errors for some hard-to-learn examples keep oscillating throughout training and never stabilize.

**Is perfect group identification necessary?** In fact, it is not necessary to perfectly identify all the subgroup examples to improve the final model's tradeoff frontier. For example, when there exists hard-to-learn majority-group examples and also easy-to-learn minority-group examples (also known as "benign bias" examples in the debiasing literature (Nam et al., 2020)), the active sampling budget is better spent in sampling some of the hard-to-learn majority examples over the easy-to-learn minority examples, so that the final tradeoff frontier is meaningfully improved both in the directions of subgroup performance and of overall accuracy. To this end, including some hard-to-learn majority examples (detected by the loss-based procedure) into introspective training help us to achieve that. This is exactly the case in the toxicity detection experiments (Section 4). As shown in Table 2, comparing between ISP-Identity v.s. ISP-Gap (trained on true group label v.s. cross-validation estimated label), the ISP-Gap attains a significantly stronger accuracy-fairness performance while sampling less minority examples. This is likely caused by the cross-validation estimator's ability in capturing challenging, non-identity-related examples instead of the easy-to-learn and identity-related

examples, which resulted in less sampling redundancy in the minority group (i.e., the identity-related comments), and led to improved tradeoff frontier for the final model.

### B.3 HYPERPARAMETERS AND COMPUTATIONAL COMPLEXITY

**Hyper-parameters** The full ISP procedure contains 3 hyper-parameters: The (optional) *cross-validated self-play* in Stage I contains all three hyper-parameters: (1) the number of ensemble models $K$ and (2) the number of examples $n_{cv}$ to train each model. Both are standard to the bootstrap ensemble procedure, and we set them to $K = 10$ and $n_{cv} = n/K$ in this work to ensure the total computation complexity is comparable to training a single model on the full dataset. (3) the early stopping criteria $\varepsilon$ for noise estimation (as discussed in the previous section B.2), we set it heuristically to $\varepsilon = 0.1$ in this work after visual inspection of the validation learning curves. The *introspective training* in Stage II does not contain additional hyperparameter other than the standard supervised learning parameters (e.g., learning rate and training epochs). We set these parameters based on a standard supervised learning hyperparameter sweep based on the full data.

**Computation Complexity** When the group annotation is available, the computation complexity of the ISP procedure (i.e., Stage II only) should be equivalent to the standard ERM procedure. On the other hand, the computational complexity of the full ISP procedure (Stage I + II) should be comparable to that of a standard two-stage debiasing method that trains multiple single models on the full dataset (Utama et al., 2020b; Liu et al., 2021; Yaghoobzadeh et al., 2021; Nam et al., 2020; Creager et al., 2021; Kim et al., 2022).

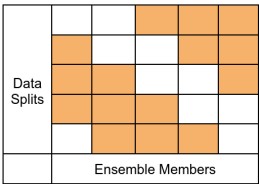

Figure 5: An example of 5-fold cross-validated ensemble. Each ensemble member received 60% of the data (highlighted in orange), and each data split receives 3 in-sample $f_{in,k}$ and 2 out-of-sample predictions $f_{out,k}$.

More specifically, The complexity of the cross-validated ensemble approach (i.e., the cross-validated self-play) is identical to that of a single model training on full data (in the big O sense). This is because for cross-validated ensembles, each ensemble member model is trained on different subsets of the data. Therefore, if the complexity for a single model is $T$, then the complexity of the cross-validated ensemble is $K * p * T$, where $p < 1$ is the percentage of sub-sampled training examples for each ensemble member, and K is the ensemble size, notice that $K * p * T = O(T)$ since $p$ and $K$ are both constants that don't scale with data size.

In practice, we find setting $p = 1/K$ (i.e., train K model on K non-overlap data splits) works rather well and has the benefit of improving the cross-validation error estimate by reducing cross-split correlation (Blum et al., 1999). This leads the complexity of cross-validated ensembles to be equal to $T$, the complexity of training a single model on the full data. Furthermore, since ensemble training is embarrassingly parallel, the complexity can also be massively reduced by leveraging multi-core machines to perform parallel training.

## C ADDITIONAL THEORY DISCUSSION

### C.1 PROPOSITION 1: WHEN WILL IT BREAK DOWN?

It is important to consider the situations where the guarantee in Proposition 1 can break down. First note that since Proposition 1 is a consistency result, it expects reasonable number of training examples in the neighborhood of $\mathbf{x}$ to ensure the convergence of $p(\mathbf{x}|b)$ (Equation (5)). In practice, the convergence is not guaranteed for the tail groups that have extraordinarily low example count in the dataset, since they can be statistically indistinguishable from the label noise under a finite-size dataset. To this end, Figure 2 provides us an empirical view of the model's bias estimation performance for the low-prevalence groups. As shown, when the tail-group examples are well-separated from the majority groups (i.e., the bias learning task is easy), a very small amount of examples is sufficient to lead to reasonable underrepresentation prediction performance (Figure 2h, only 2 examples per tail

group). This suggests that the actual number of examples required to attain reasonable performance depends on the difficulty of the bias learning task. For example, for a challenging tail group that shares much surface-form similarity with the majority groups (e.g., in Proposition 1, the tail examples are extremely close or even nested within the majority groups), more examples are expected to obtain a reasonable performance in underrepresentation prediction. Also note that the learning quality of the model representation $h(\cdot)$ depends on the quality of the underrepresentation label $b$. In particular, the noise in the underrepresentation labels can negatively impact introspective training by diminishing the log likelihood ratio $|log\,p(\mathbf{x}|b=1) - log\,p(\mathbf{x}|b=0)|$ in (5), leading the $\beta_b^\top h(\mathbf{x})$ to converge to a target of lower magnitude, hampering its ability in *bias-awareness*. This issue becomes relevant when the underrepresentation labels need to be estimated from data, where the vulnerability of fairness procedures to label noise has been well-noted in the literature (e.g., Lahoti et al. (2020)), highlighting the importance of label noise control when estimating the membership of the underrepresented groups. For example, we note that some of the now popular method (e.g., *JTT* (Liu et al., 2021)) estimates the membership of the underpresented groups using generalization error, which may be susceptible to label noise (c.f., the noise-bias-variance decomposition in Appendix A.2). We develop bias estimation procedures that controls for label noise in Section 2.2.

## C.2 ON UNDERREPRESENTED GROUP $\mathscr{B}$ IN THEOREM 1

In the main paper (section 3), we characterized the optimal allocation under assumptions on decay rates of group-specific risk. We give some further analysis and interpretation of this result here. To simplify things, we assume that there are only $\mathscr{G} = 1, 2$ groups so that the optimal allocation is characterized by a single number $\alpha \in [0, 1]$ representing the allocation to group 1 (the other group allocation would be $1 - \alpha$). We further assume that $\gamma_1 = \gamma_2 = \frac{1}{2}$ and that $c_2 = tc_1$ for some $t > 0$, From the proof of theorem Theorem 1 from section I, we know that

$$\gamma_1 \theta_1 + \gamma_2 \theta_2 = 1 \implies \theta_2 = 2 - \theta_1$$

so that we can set $\theta_1 = \theta, \theta_2 = 2 - \theta$. Then, we have that

$$\alpha^* = \frac{(\theta)^{\frac{1}{p+1}}}{(\theta)^{\frac{1}{p+1}} + (t(2-\theta))^{\frac{1}{p+1}}}$$

Further, from the proof of the theorem, we can infer that the optimal $\theta$ is

$$\begin{cases} \omega & \text{if } \omega < \frac{2}{1+t^{\frac{1}{p}}} \\ 2 - \omega & \text{if } \omega > \frac{2}{1+t^{\frac{1}{p}}} \\ \frac{2}{1+t^{\frac{1}{p}}} & \text{if } \omega = \frac{2}{1+t^{\frac{1}{p}}} \end{cases}$$

so that the optimal allocation is

$$\begin{cases} \frac{1}{1+\left(t\left(\frac{2-\omega}{\omega}\right)\right)^{\frac{1}{p+1}}} & \text{if } \omega < \frac{2}{1+t^{\frac{1}{p}}} \quad (\mathscr{B} = \{2\}) \\ \frac{1}{1+\left(t\left(\frac{\omega}{2-\omega}\right)\right)^{\frac{1}{p+1}}} & \text{if } \omega > \frac{2}{1+t^{\frac{1}{p}}} \quad (\mathscr{B} = \{1\}) \\ \frac{1}{1+t} & \text{if } \omega = \frac{2}{1+t^{\frac{1}{p}}} \quad (\mathscr{B} = \{1, 2\}) \end{cases}$$

This shows that the optimal allocation to group 1 decays as a function of $t$, the difficulty of learning group 2 relative to group 1. Further, it shows that this decay is more stark for smaller values of $\omega$ that weight the fairness risk stronger than the population level risk.

Further, group 1 belongs to the underrepresented group $\mathscr{B}$ if $\omega \geq \frac{2}{1+t^{\frac{1}{p}}}$, showing that the chances that group 1 belongs to the underrepresented group increase as $\omega$ increases (there is greater emphasis on the fairness risk) or as $t$ decreases (group 1 becomes harder to learn relative to group 2).

# D RELATED WORK

## D.1 SUPERVISED AND SEMI-SUPERVISED LEARNING UNDER DATASET BIAS

In recent years, there has been significant interest in studying robust generalization for long-tail population subgroups under dataset bias. The literature is vast and encompasses topics including fairness, debiasing, long-tail recognition, spurious correlation, distributional (i.e., domain or subpopulation) shift, etc. In the following, we focus on notable and recent work that is highly relevant to the ISP approach, and refer to works such as Caton & Haas (2020); Mehrabi et al. (2021); Hort et al. (2022) for an exhaustive survey.

Majority of the fairness and debiasing work focuses on the supervised learning setting, where the model only have access to a fixed and imbalanced dataset. Among them, the earlier work operated under the assumption that the source of dataset bias is completely known, and the group annotation is available for every training example. Then these group information is use to train a robust model by modify components of the training pipeline (e.g., training objective, regularization method, or composition of training data). For example, Levy et al. (2020); Sagawa et al. (2019); Zhang et al. (2020) proposes minimizing the worst-group loss via Distributionally Robust Optimization (DRO), which is shown to be equivalent to ERM training on a well-curated training set in some settings (Słowik & Bottou, 2022); IRM (Arjovsky et al., 2019) proposes learning a classifier that is simultaneously optimal for all groups by learning a invariant representation. Teney et al. (2021); Idrissi et al. (2022); Byrd & Lipton (2019); Xu et al. (2020) studies the effect of group-weighted loss in model's fairness-accuracy performance, and REx (Krueger et al., 2021) minimizes a combination of group-balanced and worst-case loss. Further, the recent literature has also seen sophisticated neural-network loss that modifies gradient for the tail-group examples. For example, LDAM (Cao et al., 2019) proposes to modify group-specific logits by an offset factor that is associated with group size, and equalization loss (Tan et al., 2020) uses a instance-specific mask to suppress the "discouraging gradients" from majority groups to the rare groups. On the regularization front, the examples include IRM_v1 (Arjovsky et al., 2019) that proposed to approximate the original IRM objective through gradient penalty. *Heteroskedastic Adaptive Regularization* (HAR) (Cao et al., 2020) imposes Lipschitz regularization in the neighborhood of tail-group examples. There also exists a large collection of work imposing other types of fairness constraints. Finally, the third class of methods modifies the composition of the training data by enriching the number of obsevations in the tail groups, this includes Sagawa et al. (2020); Idrissi et al. (2022) that study the impact of resampling to the worst-group performance, and Goel et al. (2020) that generates synthetic examples for the minority groups. In the setting where the group information is available, our work proposes a novel approach (introspective training) that has both a theoretical guarantee and is empirically competitive than reweighted training.

On the other hand, there exist a separate stream of work that allows for partial group annotation, i.e., the types of bias underlying a dataset is still completely known, but the group annotation is only available for a subset of the data. Most work along this direction employs semi-supervised learning techniques (e.g, confidence-threshold-based pseudo labeling), with examples include Spread Spurious Attribute (SSA) (Nam et al., 2020), BARACK (Sohoni et al., 2021) and Fair-PG (Jung et al., 2022). This setting can be considered as a special case of ISP where we use group information as the underrepresentation label to train the $p(b|\mathbf{x})$ predictor. However, our goal is distinct that we study the efficacy of this signal as an active learning policy, and also investigate its extension in the case where the label information is completely unobserved in the experiments Section 4.2.

## D.2 ESTIMATING DATASET BIAS FOR MODEL DEBIASING

In the situation where the source of dataset bias is not known and the group annotation is unavailable, several techniques has been proposed to estimate proxy bias labels for the downstream debiasing procedures. These methods roughly fall into three camps: clustering, adversarial search, and using the generalization error from a biased model.

For clustering, GEORGE (Sohoni et al., 2020) and CNC (Zhang et al., 2021) proposed estimating group memberships of examples based on clustering the last hidden-layer output. For adversarial search, REPAIR (Li & Vasconcelos, 2019), ARL (Lahoti et al., 2020), EIIL (Creager et al., 2021), BPF (Martinez et al., 2021), FAIR (Petrović et al., 2022), Prepend (Tosh & Hsu, 2022) infers the group assignments by finding the worst-case group assignments that maximize certain objective

function. For example, *Environment Inference for Invariant Learning* (EIIL) (Creager et al., 2021) infer the group membership by maximizing the gradient-based regularizer from IRM_v1.

Estimating bias label using the error from a biased model is by far the most popular technique. These include *forgettable examples* (Yaghoobzadeh et al., 2021), *Product of Experts* (PoE) (Clark et al., 2019; Sanh et al., 2020), DRiFt (He et al., 2019) and *Confidence Regularization* (CR) (Utama et al., 2020b;a) that uses errors from a separate class of weak models that is different from the main model; *Neutralization for Fairness* (RNF) (Du et al., 2021) and *Learning from Failure* (LfF) that trains a bias-amplified model of the same architecture using generalized cross entropy (GCE); and *Just Train Twice (JTT)* that directly uses the error from a standard model trained from cross entropy loss.

Notably, there also exists several work that estimates bias label using ensemble techniques, this includes *Gradient Alignment* (GA) (Zhao et al., 2021) that identifies the tail-group (i.e., bias-conflicting) examples based on the agreement between two sets of epoch ensembles, *Bias-conflicting Detection* (BCD) (Lee et al., 2022) that uses the testing error of a biased deep ensemble trained with GCE, and *Learning with Biased Committee* (LWBC) uses the testing error of a bootstrap ensemble.

To this end, our work proposes a novel *self-play estimator* (Equation (8)) that uses bootstrap ensembles to estimate the *generalization gap* due to dataset bias. *self-play estimator* has the appealing property of better controlling for label noise while more stably estimating model variance, addressing two weaknesses of the naive predictive error estimator used in the previous works.

### D.3 REPRESENTATION LEARNING UNDER DATASET BIAS

Originated from the fairness literature, the earlier work in debiased representation learning has focused on identifying a representation that improves a model's fairness properties, as measured by notions such as demographic parity (DP), equalized odds (EO), or sufficiency (Arjovsky et al., 2019; Arjovsky, 2020; Creager et al., 2021; Shui et al., 2022). This is commonly achieved by striving to learn a representation that is invariant with respect to the group information. Such methods are commonly characterized as *in-processing methods* in the existing survey of the fairness literature (Caton & Haas, 2020; Hort et al., 2022), and was categorized into classes of approaches including adversarial training (Beutel et al., 2017; Kim et al., 2019; Ragonesi et al., 2021; Zhu et al., 2021; Madras et al., 2018), regularization (Bahng et al., 2020; Tartaglione et al., 2021; Arjovsky et al., 2019), contrastive learning (Shen et al., 2021; Park et al., 2022; Cheng et al., 2020) and its conditional variants (Gupta et al., 2021; Tsai et al., 2021a;b; Chi et al., 2022), or explicit solutions to a bi-level optimization problem (Shui et al., 2022). (Please see Caton & Haas (2020); Hort et al. (2022) for a complete survey). However, some later works questions the necessity and the sufficiency of such approaches. For example, some work shows that careful training of the output head along is sufficient to yield improved performance in fairness and bias mitigation (Kang et al., 2019; Du et al., 2021; Kirichenko et al., 2022), and Cherepanova et al. (2021) shows that models with fair feature representations do not necessarily yield fair model behavior.

At the meantime, a separate stream of work explores the opposite direction of encouraging the model to learn diverse hidden features. For example, Locatello et al. (2019b;a) establish a connection between the notion of feature disentanglement and fairness criteria, showing that feature disentanglement techniques may be a useful property to encourage model fairness when sensitive variables are not observed. However, such techniques often involves specialized models (e.g., VAE) which restricts the broad applicability of such approaches. Some other work explores feature augmentation techniques to learn both invariant and spurious attributes, and use them to debias the output head (Lee et al., 2021). Finally, a promising line of research has been focusing on using self-supervised learning to help the model avoid using spurious features in model predictions (Chen et al., 2020b; Xie et al., 2020; Cai et al., 2021; Hamidieh et al., 2022). Our work follows this latter line of work by proposing novel techniques to encourage model to learn diverse features that is *bias-aware*, but with a distinct purpose of better uncertainty quantification.

### D.4 ACTIVE LEARNING UNDER DATASET BIAS

In recent years, the role of training data in ensuring the model's fairness and bias-mitigation performance has been increasing noticed. Notably, (Chen et al., 2018) presented some of the earlier theoretical and empirical evidence that increasing training set size along is already effective in mitigating model unfairness. Correspondingly, under the assumption that the *group information in the unlabelled set is fully known*, there has been several works that studies group-based sampling strate-

gies and their impact on model behavior. For example, Rai et al. (2010); Wang et al. (2014) shows group-based active sampling strategy improves model performance under domain and distributional shifts, and Abernethy et al. (2022) proves a guarantee for a worst-group active sampling strategy's ability in helping the SGD-trained model to convergence to a solution that attains min-max fairness. A second line of research focuses on designing better active learning objectives that incorporates fairness constraints, e.g., *Fair Active Learning (FAL)* (Anahideh et al., 2022) and *PANDA* (Sharaf et al., 2022). Agarwal et al. (2022) introduce a data repair algorithm using the coefficient of variation to curate fair and contextually balanced data for a protected class(es). Furthermore, there exists few active learning works formulating the objective of their method as optimizing a fairness-aware objective. For example, *Slice Tuner* (Tae & Whang, 2021) proposes adaptive sampling strategy based on per-group learning curve to minimize fairness tradeoff, performs numeric optimization. Cai et al. (2022) which formalized the fairness learning problem as an min-max optimization objective, however their did not conduct further theoretical analysis of their objective, but instead proposed a per-group sampling algorithm based predicted model error using linear regression. Finally, a recent line of active learning work has been well-developed to formulate active learning from the perspective of distributional matching between labeled and unlabeled datasets. (Shui et al., 2020; de Mathelin et al., 2021; Mahmood et al., 2021; Xie et al., 2022). These approaches are well-equipped to hand distributional shift, but often comes with the cost of assuming knowledge of the test feature distribution at the training time (Shui et al., 2020; de Mathelin et al., 2021; Mahmood et al., 2021; Xie et al., 2022). In comparison, our work conducts theoretical analysis of the optimization problem Section 3, and our proposed method (ISP) does not require a priori knowledge (e.g., group information) from the unlabelled set.

On the other hand, there exists active re-sampling methods that do not require the knowledge of group information in the unlabelled set. For example, Amini et al. (2019) learns the data distribution using a VAE model under additional supervision of class / attribute labels, and then perform IPW sampling with respect to learned model. REPAIR (Li & Vasconcelos, 2019) that estimates dataset bias using prediction error of a weak model, and then re-train model via e.g., sample re-weighting based on the estimated bias. The bias estimation method used in this work is analogous to that of the JTT, which we compare with in our work. A work close to our direction is Branchaud-Charron et al. (2021), which shows DNN uncertainty (i.e., BatchBALD with Monte Carlo Dropout (Kirsch et al., 2019)) helps the model to achieve fairness objectives in active learning on a synthetic vision problem. Our empirical result confirms the finding of Branchaud-Charron et al. (2021) on realistic datasets, and we further propose techniques to improve the vanilla DNN uncertainty estimators for more effective active learning under dataset bias.

As an aside, a recent work Farquhar et al. (2020) studies the statistical bias in the estimation of active learning objectives due to the non-i.i.d. nature of active sampling. This is separate from the issue of dataset bias (i.e., imbalance in data group distribution) which we focus on in this work.

### D.5 UNCERTAINTY ESTIMATION WITH DNNS

In recent years, the probabilistic machine learning (ML) literature has seen a plethora of work that study enabling calibrated predictive uncertainty in DNNss. Given a model $f$, the probabilistic DNN model aims to learn a predictive distribution for the model function $f$, such that given training data $D = \{(y_i, \mathbf{x}_i)\}_{i=1}^n$ and a testing point $\mathbf{x}_{test}$, the model outputs a predictive distribution $f(\mathbf{x}_{test}) \sim P(f|\mathbf{x}_{test}, D)$ rather than a simple point prediction. To this end, the key challenge is to learn a predictive distribution (implicitly or explicitly) during the SGD-based training process of DNN, generating calibrated predictive uncertainty without significantly impacting the accuracy or latency when compared to a deterministic DNN.

To this end, the classic works focus on the study of Bayesian neural networks (BNNs) (Neal, 2012), which took a full Bayesian approach by *explicitly* placing priors to the hidden weights of the neural network, and performance MCMC or variance inference during learning. Although theoretically sound, BNN are delicate to apply in practice, with its performance highly dependent on prior choice and inference algorithm, and are observed to lead to suboptimal predictive accuracy or even poor uncertainty performance (e.g., under distributional distribution shift) (Wenzel et al., 2020a; Izmailov et al., 2021). Although there exists ongoing works that actively advancing the BNN practice (e.g., Dusenberry et al. (2020)). On the other hand, some recent work studies computationally more approaches that *implicitly* learn a predictive distribution as part of deterministic SGD training. Notable examples include Monte Carlo Dropout (Gal & Ghahramani, 2016) which generates predictive

distribution by enabling the random Dropout mask during inference, and ensemble approaches such as Deep Ensemble (Lakshminarayanan et al., 2017) and their later variants (Maddox et al., 2019; Wenzel et al., 2020b; Havasi et al., 2020) that trains multiple randomly-initialized networks to learn the modes of the posterior distribution of the neural network weights (Wilson & Izmailov, 2020). Although generally regarded as the state-of-the-art in deep uncertainty quantification, these methods are still computationally expensive, requiring multiple DNN forward passes at the inference time.

At the meantime, a more recent line of research avoids probabilistic inference for the hidden weights altogether, focusing on learning a scalable probabilistic model (e.g., Gaussian process) to replace the last dense layer of the neural network (Van Amersfoort et al., 2020; van Amersfoort et al., 2021; Liu et al., 2022; Collier et al., 2021). A key important observation in this line of work is the role of hidden representation quality in a model's ability in obtaining high-quality predictive uncertainty. In particular, Liu et al. (2022); Van Amersfoort et al. (2020) suggests that this failure mode in DNN uncertainty can be caused by an issue in representation learning known as *feature collapse*, where the DNN over-focuses on correlational features that help to distinguish between output classes on the training data, but ignore the non-predictive but semantically meaningful input features that are important for uncertainty quantification. (Ming et al., 2022) also observed that DNN exhibits particular modes of failure in out-of-domain (OOD) detection in the presence of dataset bias. Later, Tran et al. (2022); Minderer et al. (2021) suggests that this issue can be partially mitigated by large-scale pre-traininig with large DNNs, where larger pre-trained DNN's tend to exhibit stronger uncertainty performance even under spurious correlation and subpopulational shift. In this work, we confirm this observation in the setting of dataset bias in Figure 2), and propose simple procedures to mitigate this failure mode in representation learning without needing any change to the DNN model, and illustrates improvement even on top of large-scale pre-trained DNNs (BERT).

**Deep uncertainty methods in active learning.** Active learning with DNNs is an active field with numerous theoretical and applied works, we refer to Matsushita et al. (2018); Ren et al. (2021) for comprehensive survey, and only mention here few notable methods that involves DNN uncertainty estimation techniques. Under a classification model, the most classic approach to uncertainty-based active learning is to use the predictive distribution's entropy, confidence or margin as the acquisition policy (Settles, 1994). Notice that in the binary classification setting, these three acquisition policy are rank-equivalent since they are monotonic to the distance between $max[p(\mathbf{x}), 1 - p(\mathbf{x})]$ and the null probability value of 0.5. On the other hand, *Batch Active learning by Diverse Gradient Embeddings (BADGE)* (Ash et al., 2019) proposes to blend diversity-based acquisition policy into uncertainty-based active learning, by applying k-means++ algorithm to the gradient embedding of the class-specific logits (which quantifies uncertainty). As a result, *BADGE* may also suffer from the pathology in model representation under dataset bias, which this work is attempt to address.

Finally, Houlsby et al. (2011) has proposed a information-theoretic policy Bayesian active learning by disagreement (BALD), which measures the mutual information between data points and model parameters and is adopted in the deep uncertainty literature (Gal et al., 2017; Kirsch et al., 2019; Kothawade et al., 2022). However, stable estimation of mutual information can be delicate in practice, and we leave the investigation of these advanced acquisition policies under dataset bias for future work.

# E    EXPERIMENT DETAILS AND FURTHER DISCUSSION

## E.1    2D CLASSIFICATION

We train a 10-member neural Gaussian process ensemble (as introduced in Appendix A.4), where each ensemble member is based on a 6-layer Dense residual network with 512 hidden units and pre-activation dropout mask (rate = 0.1). The model is trained using Adam optimizer ( learning rate = 0.1) under cross entropy loss, and with a batch size 512 for 100 epochs. After training, each ensemble member returns a tuple of predicted label probability, predicted under-representation probability and predictive uncertainty $\{(p_k(y|\mathbf{x}), p_k(b|\mathbf{x}), v_k(\mathbf{x}))\}_{k=1}^{10}$, and we compute the ensemble's predicted probability surface as $\mathbb{E}_k[p(y|\mathbf{x})]$, predicted underrepresentation surface as $\mathbb{E}_k[p_k(b|\mathbf{x})]$, and the predictive uncertainty surface as $\mathbb{E}_k[v_k(y|\mathbf{x})]$, where $\mathbb{E}_k$ is the empirical average over the ensemble member predictions. The predictive uncertainty surface of individual members is shown in Figures 6-7.

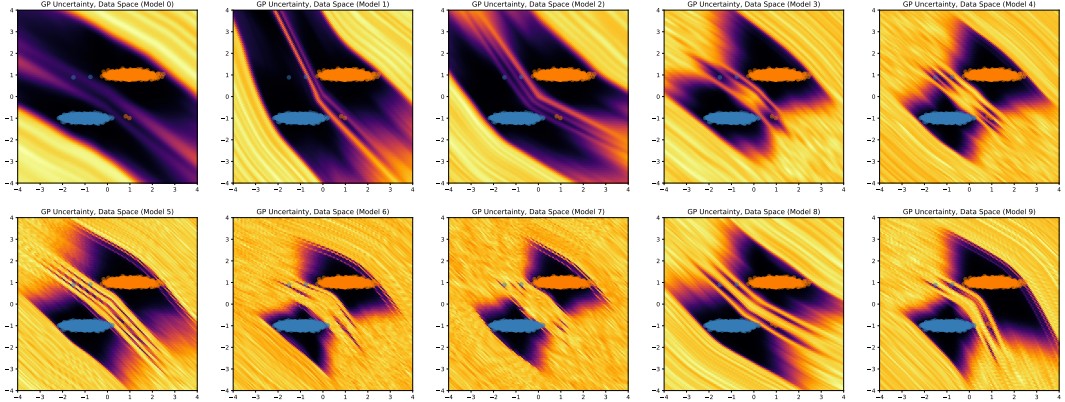

Figure 6: Uncertainty surface of individual ensemble members, ERM training

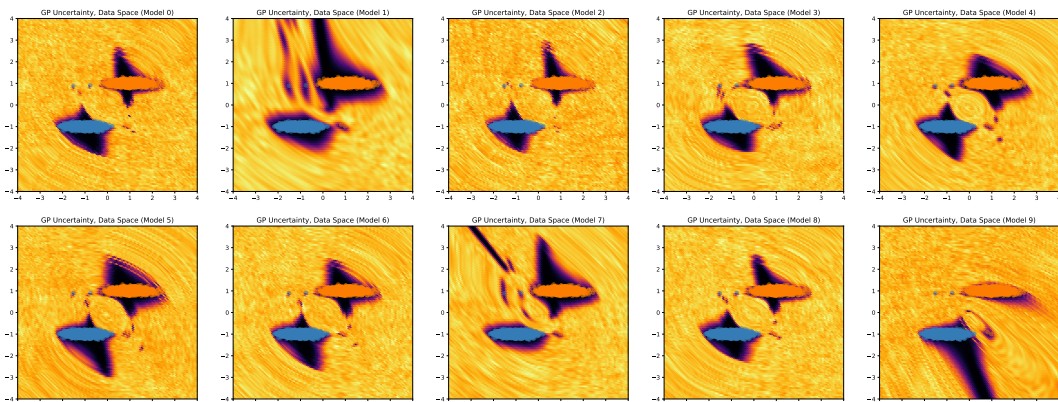

Figure 7: Uncertainty surface of individual ensemble members, introspective training.

As shown, compared to the ERM-trained model, the introspective-trained model generates similar label prediction decision $I(p(y|\mathbf{x}) > 0.5)$ (Figures 2a v.s. 2e), but with much improved uncertainty surface (Figures 2b v.s. 2f). Specifically, we compute predictive variance using the standard Gaussian process variance formula $v(\mathbf{x}_{test}) = \mathbf{k}(\mathbf{x}_{test})^\top \mathbf{V} \mathbf{k}(\mathbf{x}_{test})$, where $\mathbf{k}(\mathbf{x}_{test}) = [k(\mathbf{x}_{test}, \mathbf{x}_1), \ldots, k(\mathbf{x}_{test}, \mathbf{x}_n)]_{n \times 1}$ is a vector of kernel distances based on the embedding distances $||h(\mathbf{x}_{test}) - h(\mathbf{x}_i)||_2$ from the training data (Appendix A.4). As shown, the model uncertainty under ERM model are not sufficiently sensitive to directions in the data space that are irrelevant for making prediction decisions on the training data (i.e., the directions that are parallel to the decision boundary) (Figure 2f). As a result, it did not learn sufficiently diverse hidden features, leading to a significantly warped representation space that is extremely stretched out in the direction that is orthogonal to the decision boundary, and extremely compressed otherwise (Figure 2g). Consequently, the model cannot strongly distinguish the minority examples from the majority examples in the representation space, and can become overconfident even in unseen regions that was never covered by training data. This can be undesirable for uncertainty quantification under data bias, especially for the purpose of identifying underrepresented minority examples, where the distinguishing features between the minority and the majority examples are not predictive for the target label (e.g., the image background). This issue is further exacerbated in the single models (see Figure 6). In comparison, the uncertainty surface from an introspective-trained model does not suffer from this failure case. As shown in Figure 2b, the model is less inclined to become overconfident in unseen regions, especially in the neighborhood of the minority examples. Correspondingly in the representation space, the model learned more diverse features and is able to better distinguish the minority examples from the majority examples (Figure 2b). To understand how introspective training induces such improvement in model behavior, Figures (2g) and (2h) visualize the model's underrepresentation prediction $p(b|\mathbf{x})$ in the representation space and the data space, respectively. As shown, due to the need of predicting the underrepresented examples (i.e., "introspection") during training, the model has to learn hidden features that distinguishes the minority examples

from the majority examples in its representation space, to a degree that they can be separated by a linear decision boundary in the last layer (Figure 2h). Consequently, the model naturally learns a more disentangled representation space through simple multi-task training, and is able to provide predicted bias probabilities $p(b|\mathbf{x})$ (Figure 2h) in addition to high-quality predictive uncertainty (Figure 2b) for the downstream active learning applications.

### E.2    TABULAR AND LANGUAGE EXPERIMENTS

**Data.**    For tabular data, we use the U.S. Census Income data `adult` from the official UCI repository[4]. For the language task, we use the `CivilCommentsIdentity` from the TensorFlow Dataset repository[5]. For Census Income, we define the underrepresented groups as the union of (Female, High Income) and (Black, High Income); for Toxicity Detection, we define the underrepresented groups as the identity $\times$ label combination (male, female, white, black, LGBTQ, christian, muslim, other religion) $\times$ (toxic, non-toxic) (16 groups in total) as in (Koh et al., 2021). For CivilComments, the identity annotation is a value between $(0, 1)$ (it is the average rating among multiple raters), and we include an example into the underrepresented group only if the rating $> 0.99$ (i.e. all raters agree about the identity) following (Koh et al., 2021). However, we do note that this leads to a under coverage of the group membership, as many comments with plausible identity mentions are not included into the group identity labels.

**Model.**    For tabular experiments, we use a 2-layer Dense ResNet model with 128 hidden units and pre-activation dropout rate = 0.1, using a random-feature Gaussian process with hidden dimension 256 as the output layer (Liu et al., 2022) (In the preliminary experiments, we tried larger models with update to 6-layers and 1024 hidden units, and did not observe significant improvement). For language experiments, we used $\text{BERT}_{\text{small}}$ mode initialized from the official pre-trained checkpoint released at BERT GitHub page(Turc et al., 2019)[6]. In each active learning round, we train the Dense ResNet model with Adam optimizer with learning rate 0.1, batch size 256 and maximum epoch 200; and train the BERT model with AdamW optimizer (learning rate 1e-5) for 6 epochs with batch size 16.

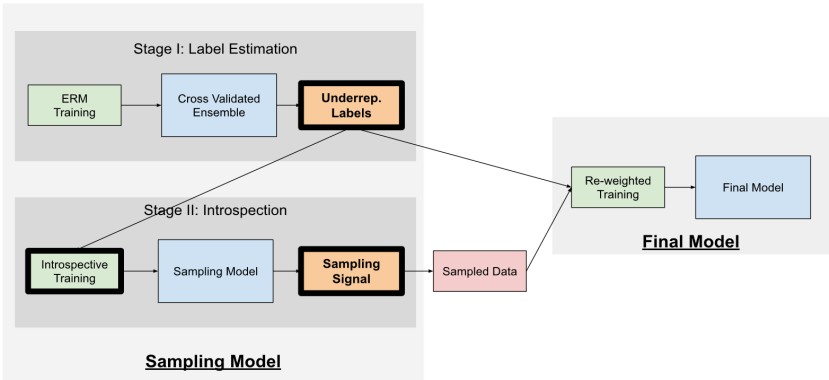

Figure 8: Experiment Protocol. Boxes with thick outlines (Underrepresentation Label, Introspective Training Method, Sampling Signal) indicates the experiment components where the methods differ.

**Active Learning Protocol.**    Figure 8 visualizes the experiment protocol. As shown, in each stage, we first (optionally) trains a cross validated ensemble to estimate the under-representation labels, where we split the data into 10 cross-validation splits, and train ensemble members on 1 split and predict the rest of the 9 splits. We then use the ensemble's in-sample and out-of-sample predictions to compute the underrepresentation label $\hat{b}_i$ (Equation (8)), and conduct introspective training (eq. (4)) to generate the final active sampling signals for 8 rounds to generate the final sampled data (red box). At the end of round 8, we estimate the underrepresentation label for the final sampled data, and send it to the final model for reweighted training to generate the full accuracy-fairness frontier. The sampling model is always a 10-member ensemble of neural Gaussian process (introduced in Appendix A.4),

---

[4]https://archive.ics.uci.edu/ml/datasets/adult
[5]https://www.tensorflow.org/datasets/catalog/civil_comments
[6]https://github.com/google-research/bert

and the final model is always a single DNN with architecture identical to the sampling model (i.e., 2-layer Dense ResNet for census income and BERT$_{small}$ for toxicity detection).

For both tasks, we randomly sample as small subset as the initial labelled dataset (2,500 out of 32,561 total training examples for census income, and 50,000 out of total 405,130 examples for toxicity detection), and use the rest of the training set as the unlabelled set for active learning. For each sampling round, the AL model acquires 1,500 examples for census income, and 15,000 examples for the toxicity detection, so the total sample reaches roughly half of the total training set size after 8 rounds.

In the final model training, we use the standard re-weighting objective (Liu et al., 2021):

$$\sum_{(x,y)\notin \hat{\mathcal{B}}} L_{ce}(y, f(\mathbf{x})) + \lambda \sum_{(x,y)\in \hat{\mathcal{B}}} L_{ce}(y, f(\mathbf{x}))$$

where $\hat{\mathcal{B}}$ is the set of underrepresented examples identified by the underrepresentation label, i.e., $(x_i, y_i) \in \hat{\mathcal{B}}$ if $\hat{b}_i > t$. We vary the thresholds $t$ and the up-weight coefficient $\lambda$ over a 2D grid ($t \in \{0.05, 0.1, 0.15, ..., 1.0\}$ and $log(\lambda) \in \{0., 0.5, 1, 1.5, ..., 10.\}$) to get a collection of model accuracy-fairness performances (i.e., accuracy v.s. worst-group accuracy), and use them to identify the Pareto frontier defined by this combination of data and reweighting signal.

**Active Learning Signals.** In this work, we consider four types of active sampling signals. Recall that the sampling model (neural Gaussian process ensemble) is a K-member ensemble that generates three predictive quantities: (1) label probability $\{p_k(y|\mathbf{x})\}_{k=1}^{10}$, (2) underrepresentation probability $\{p_k(b|\mathbf{x})\}_{k=1}^{10}$ and (3) predictive variance $\{v(\mathbf{x})\}_{k=1}^{10}$ (recall that $\mathbb{E}_k$ and $Var_k$ are the empirical mean and variance).

- **Margin**: The gap between the highest class probability and the second highest class probability for the output label. In the binary prediction context, this is equivalent to $2 * |p(y|\mathbf{x}) - 0.5|$, i.e., the gap between the mean predicted probability and the null value of 0.5. We use the mean predictive label probability of the ensemble, which leads to:

$$Margin(\mathbf{x}) = 2 * |\mathbb{E}_k(p_k(y|\mathbf{x})) - 0.5|.$$

- **Predicted Underrepresentation**: The mean predictive underrepresentation probability of the ensemble, which leads to:

$$Underrep(\mathbf{x}) = \mathbb{E}_k(p_k(b|\mathbf{x})).$$

- **Diversity**: i.e., Ensemble Diversity (introduced in Appendix A.4). The variance of label predictions:

$$Diversity(\mathbf{x}) = Var_k(p_k(y|\mathbf{x})).$$

- **Variance**: i.e., Predictive Variance (introduced in Appendix A.4). The mean of predictive variances:

$$Variance(\mathbf{x}) = \mathbb{E}_k(v_k(\mathbf{x})).$$

## F    NOISE-BIAS-VARIANCE DECOMPOSITION IN RIDGE REGRESSION

Consider fitting a ridge regression model $f(\mathbf{x}_i) = \beta^\top \mathbf{x}_i$ to the Gaussian observation data $y_i = \theta^\top \mathbf{x}_i + \varepsilon, \varepsilon \sim N(0, \sigma^2)$ under an imbalanced experiment design, where we have $|\mathcal{G}|$ treatment groups and $n_g$ observations in each group. . Here, $\mathbf{x}_i = [1_{g_i=1}, ..., 1_{g_i=|\mathcal{G}|}]$ is a $|\mathcal{G}| \times 1$ one-hot indicator of the membership of $g_i$ for each group in $\mathcal{G}$, and $\theta = [\theta_1, ..., \theta_{|\mathcal{G}|}]$ is the true effect for each group. Then, under ridge regression, the noise-bias-variance decomposition for group $g$ is $E_D(L(y, f_D)) = \sigma^2 + \frac{(\lambda \theta_g)^2}{(n_g + \lambda)^2} + \frac{\sigma^2 n_g}{(n_g + \lambda)^2}$, where the regularization parameter $\lambda$ modulates a trade-off between the *bias* and *variance* terms. In Appendix F.3, we also treat the case of group-specific noise $\varepsilon_i \overset{indep}{\sim} N(0, \sigma_g^2)$ .

### F.1    ERROR DECOMPOSITION IN A GENERAL SETTING

We first derive the decomposition in a general setting with data $\{y_i, \phi_i\}_{i=1}^n$, where $\phi_i$ is the $d \times 1$ (fixed) features that follows a distribution $P(\phi)$. We consider a well-specified scenario where the data

generation mechanism as:

$$y_i = \tilde{y}_i + \varepsilon, \quad \text{where} \quad \tilde{y}_i = \theta^\top \phi_i, \ \varepsilon \overset{iid}{\sim} N(0, \sigma^2),$$

and $\theta_{d \times 1} = [\theta_1, \ldots, \theta_{|\mathscr{G}|}]$ is the true coefficient. Under ridge regression, we fit a linear model $f(\mathbf{x}_i) = \phi_i^\top \beta$ to the data by minimizing the following squared loss objective:

$$||\mathbf{y}_{n \times 1} - \Phi_{n \times d} \beta_{d \times 1}||_2^2 + \lambda ||\beta||_2^2,$$

which gives rise to the following solution:

$$\hat{\beta} = (\Phi^\top \Phi + \lambda I_d)^{-1} \Phi^\top \mathbf{y}. \tag{13}$$

notice $\hat{\beta}$ is a random variable that depends on the data $\Phi_{n \times d} = [\phi_1^\top, \ldots, \phi_n^\top] \overset{iid}{\sim} P(\phi)$. Notice that under squared loss, the ensemble predictors $\bar{f} = argmin_f E_\Phi[(f - \hat{\beta}^\top \mathbf{x}_i)^2]$ is simply the mean of individual predictors, i.e., $\bar{f} = E_\Phi(\hat{\beta}^\top \mathbf{x}_i) = \bar{\beta}^\top \mathbf{x}_i$, where $\bar{\beta} = E_\Phi(\hat{\beta})$.

Consequently, given a new observation $\{y, \phi\}$, the noise-bias-variance decomposition of $\hat{\beta}$ under squared loss is:

$$\begin{aligned}
E[(y - \Phi\hat{\beta})^2] &= E_y[(y - \tilde{y})^2] + (\tilde{y} - \bar{\beta}^\top \phi_i)^2 + E_\Phi[\bar{\beta}^\top \phi_i - \hat{\beta}^\top \phi_i]^2 \\
&= \underbrace{\sigma^2}_{Noise} + \underbrace{\phi_i^\top [\theta - \bar{\beta}][\theta - \bar{\beta}]^\top \phi_i}_{Bias} + \underbrace{\phi_i^\top Var(\hat{\beta}) \phi_i}_{variance}.
\end{aligned} \tag{14}$$

As shown, to obtain a closed-form expression of the decomposition, we need to first derive the expressions of $Bias(\hat{\beta}) = [\theta - \bar{\beta}]$ and $Var(\hat{\beta})$. Under the expression of the ridge predictor (13), we have:

$$\begin{aligned}
Bias(\hat{\beta}) &= [\theta - \bar{\beta}] \\
&= \theta - E[(\Phi^\top \Phi + \lambda I_d)^{-1} \Phi^\top \Phi \theta] \\
&= E[I - (\Phi^\top \Phi + \lambda I_d)^{-1} \Phi^\top \Phi] \theta \\
&= \lambda * E[(\Phi^\top \Phi + \lambda I_d)^{-1}] \theta; \\
Var(\hat{\beta}) &= E(Var(\hat{\beta}|\Phi)) + Var(E(\hat{\beta}|\Phi)),
\end{aligned}$$

with

$$\begin{aligned}
E(Var(\hat{\beta}|\Phi)) &= E[(\Phi^\top \Phi + \lambda I_d)^{-1} \Phi^\top Var(\mathbf{y}) \Phi (\Phi^\top \Phi + \lambda I_d)^{-1}] \\
&= \sigma^2 E[(\Phi^\top \Phi + \lambda I_d)^{-1} \Phi^\top \Phi (\Phi^\top \Phi + \lambda I_d)^{-1}] \\
&= \sigma^2 * E[(\Phi^\top \Phi + \lambda I_d)^{-1} - \lambda (\Phi^\top \Phi + \lambda I_d)^{-2}]. \\
Var(E(\hat{\beta}|\Phi)) &= E[S\theta\theta^\top S^\top] - E[S]\theta\theta^\top E[S^\top]
\end{aligned}$$

where $S = (\Phi^\top \Phi + \lambda I_d)^{-1} \Phi^\top \Phi$.

As shown, the above expression depends on the random-matrix moments $E[(\Phi^\top \Phi + \lambda I_d)^{-1}]$, $E[(\Phi^\top \Phi + \lambda I_d)^{-2}]$, $E[(\Phi^\top \Phi + \lambda I_d)^{-1} \Phi^\top \Phi]$ and $E[S\theta\theta^\top S^\top]$.

## F.2 Error Decomposition under Orthogonal Design

The above moments are in general difficult to solve due to the involvement of matrix inverse and product within the expectation. However, a closed-form expression is possible under an orthogonal design where $\phi_i = [1_{g_i=1}, \ldots, 1_{g_i=|\mathscr{G}|}]$ is the one-hot vector of treatment group memberships. Then, denote $Diag[z_g]$ the diagonal matrix with diagonal elements $z_g$ and $[z_{gg'}]_{gg'}$ the full matrix whose

$(g, g')$ element is $z_{gg'}$, we have:

$$\Phi^\top \Phi = diag[n_g]$$

$$E[(\Phi^\top \Phi + \lambda I_d)^{-1}] = diag[\frac{1}{n_g + \lambda}],$$

$$E[(\Phi^\top \Phi + \lambda I_d)^{-2}] = diag[\frac{1}{(n_g + \lambda)^2}],$$

$$E[(\Phi^\top \Phi + \lambda I_d)^{-1} \Phi^\top \Phi] = diag[\frac{n_g}{n_g + \lambda}],$$

and

$$E[(\Phi^\top \Phi + \lambda I_d)^{-1} \Phi^\top \Phi \theta \theta^\top \Phi^\top \Phi (\Phi^\top \Phi + \lambda I_d)^{-1}] = [\frac{n_g n'_g}{(n_g + \lambda)(n'_g + \lambda)} \theta_g \theta_{g'}]_{gg'}.$$

We are now ready to derive the full decomposition (15), without loss of generality, we assume $\phi_i$ belongs to group $g$. Then:

$$\phi_i^\top Bias(\hat{\beta}) = \lambda * \phi_i^\top E[(\Phi^\top \Phi + \lambda I_d)^{-1}] \theta = \frac{\lambda}{n_g + \lambda} \theta_g;$$

$$\phi_i^\top Var(\hat{\beta}) \phi_i = \sigma^2 * \phi_i^\top E[(\Phi^\top \Phi + \lambda I_d)^{-1} - \lambda (\Phi^\top \Phi + \lambda I_d)^{-2}] \phi_i;$$

$$= \frac{\sigma^2}{n_g + \lambda} - \frac{\lambda \sigma^2}{(n_g + \lambda)^2} = \frac{\sigma^2 n_g}{(n_g + \lambda)^2}.$$

Consequently, we have the noise-bias-variance decomposition in (15) as:

$$Noise: \quad \sigma^2;$$

$$Bias: \quad ||\phi_i^\top Bias(\hat{\beta})||_2^2 = \frac{(\lambda \theta_g)^2}{(n_g + \lambda)^2};$$

$$Uncertainty: \quad \phi_i^\top Var(\hat{\beta}) \phi_i = \frac{\sigma^2 n_g}{(n_g + \lambda)^2}.$$

### F.3    ERROR DECOMPOSITION UNDER ORTHOGONAL DESIGN AND HETEROGENEOUS NOISE

We now consider the case where $y_i \sim N(\theta^\top \phi_i, \sigma_g^2)$ follows a normal distribution with group-specific noise. Using the same decomposition as in F.1, we see that:

$$E[(y - \Phi\hat{\beta})^2] = E_y[(y - \tilde{y})^2] + (\tilde{y} - \bar{\beta}^\top \phi_i)^2 + E_\Phi[\bar{\beta}^\top \phi_i - \hat{\beta}^\top \phi_i]^2$$

$$= \underbrace{\sigma_g^2}_{Noise} + \underbrace{\phi_i^\top [\theta - \bar{\beta}][\theta - \bar{\beta}]^\top \phi_i}_{Bias} + \underbrace{\phi_i^\top Var(\hat{\beta}) \phi_i}_{variance}. \quad\quad (15)$$

As shown, the nature of the bias and variance decomposition in fact does not change, and the noise component is now the group-specific variance $\sigma_g^2$. Therefore, by following the same derivation as in Appendix F.2, we have:

$$Noise: \quad \sigma_g^2;$$

$$Bias: \quad ||\phi_i^\top Bias(\hat{\beta})||_2^2 = \frac{(\lambda \theta_g)^2}{(n_g + \lambda)^2};$$

$$Uncertainty: \quad \phi_i^\top Var(\hat{\beta}) \phi_i = \frac{\sigma^2 n_g}{(n_g + \lambda)^2}.$$

# G  PROOF OF PROPOSITION 1

Through introspective training, there is a guarantee on a model's bias-awareness based on its hidden representation and uncertainty estimates. At convergence, a well-trained model $f = (f_y, f_b)$ should satisfy the property that $p(b = 1|x) = \sigma(f_b(\mathbf{x}))$.

*(I) (Bias-aware Hidden Representation)* We denote the odds for $\mathbf{x}$ belonging to the underrepresented group $\mathscr{B}$ as $o_b(\mathbf{x}) = p(\mathbf{x}|b = 1)/p(\mathbf{x}|b = 0)$. Using Bayes' theorem, we derive the following:

$$p(b|\mathbf{x}) = \sigma(\beta^T h(\mathbf{x}) + \beta_0)$$

$$log \frac{p(b = 1|\mathbf{x})}{p(b = 0|\mathbf{x})} = \beta^T h(\mathbf{x}) + \beta_0$$

$$log \frac{p(\mathbf{x}|b = 1)p(b = 1)}{p(\mathbf{x}|b = 0)p(b = 0)} = \beta^T h(\mathbf{x}) + \beta_0$$

$$\beta^T h(\mathbf{x}) + \beta_0 = log P(\mathbf{x}|b = 1) - log P(\mathbf{x}|b = 0) + log \frac{p(b = 1)}{p(b = 0)}$$

$$\beta^T h(\mathbf{x}) + \beta_0 = log\, o_b(\mathbf{x}) + log \frac{p(b = 1)}{p(b = 0)} \tag{16}$$

Hence, the hidden representation is aware of the likelihood ratio of whether an example $\mathbf{x}$ belongs to the underrepresented group, and the last-layer bias $\beta_0$ corresponds to the marginal likelihood ratio of the prevalence of the underrepresented groups $p(b = 1)/p(b = 0)$ .

*(II) (Bias-aware Embedding Distance)* Next, we examine the embedding distance between two examples $(\mathbf{x}_1, \mathbf{x}_2)$, i.e., $||h(\mathbf{x}_1) - h(\mathbf{x}_2)||_2$.

The Cauchy-Schwarz inequality states that for two vectors $\mathbf{u}$ and $\mathbf{v}$ of the Euclidean space, $|\langle \mathbf{u}, \mathbf{v} \rangle| \leq ||\mathbf{u}|| \, ||\mathbf{v}||$. Hence, the distance between two embeddings can be expressed as $\beta^T [h(\mathbf{x}_1) - h(\mathbf{x}_2)] \leq ||\beta||_2 ||h(\mathbf{x}_1) - h(\mathbf{x}_2)||_2$. Using this property and Equation (16), we derive the following:

$$\beta^T h(\mathbf{x}_1) - \beta^T h(\mathbf{x}_2) = log\, o_b(\mathbf{x}_1) - log\, o_b(\mathbf{x}_2)$$

$$\beta^T [h(\mathbf{x}_1) - h(\mathbf{x}_2)] = log\, o_b(\mathbf{x}_1) - log\, o_b(\mathbf{x}_2)$$

$$log\, o_b(\mathbf{x}_1) - log\, o_b(\mathbf{x}_2) \leq ||\beta||_2 ||h(\mathbf{x}_1) - h(\mathbf{x}_2)||_2$$

$$\frac{1}{||\beta||_2} [log\, o_b(\mathbf{x}_1) - log\, o_b(\mathbf{x}_2)] \leq ||h(\mathbf{x}_1) - h(\mathbf{x}_2)||_2$$

$$\frac{1}{||\beta||_2} log \frac{o_b(\mathbf{x}_1)}{o_b(\mathbf{x}_2)} \leq ||h(\mathbf{x}_1) - h(\mathbf{x}_2)||_2 \tag{17}$$

Since the above inequality is invariant to the relative position of $(\mathbf{x}_1, \mathbf{x}_2)$, we also have: $\frac{1}{||\beta||_2} log \frac{o_b(\mathbf{x}_2)}{o_b(\mathbf{x}_1)} \leq ||h(\mathbf{x}_1) - h(\mathbf{x}_2)||_2$, which implies:

$$||h(\mathbf{x}_1) - h(\mathbf{x}_2)||_2 \geq \frac{1}{||\beta||_2} * max(log \frac{o_b(\mathbf{x}_1)}{o_b(\mathbf{x}_2)}, log \frac{o_b(\mathbf{x}_2)}{o_b(\mathbf{x}_1)}).$$

As shown, the distance between the hidden embeddings $h(\mathbf{x}_1)$, $h(\mathbf{x}_2)$ is lower-bounded by the log-odds ratio that a given example is in the underrepresented group. With this guarantee on the model's learned embedding distance, we expect the hidden features to be more diverse than when trained on the main task alone, since it needs to sufficient features to distinguish the underrepresented-group examples from those of the majority in the hidden space.

# H  PERFORMANCE GUARANTEE FOR LOSS-BASED TAIL-GROUP DETECTION

In this section, we derive a lower bound for the group detection performance based on the self-play estimator introduced in Section 2.2. We derive the result in a general setting without further assumptions on the data distribution. The goal here is to provide a broadly applicable, and mathematically

rigorous account of how various aspects of the data distribution and the model behavior impacts the performance of the subgroup detection procedure, rather than deriving the tightest possible performance guarantee for a specific family of model class or data distribution. Our proof technique develops a novel Cantelli-type lower bound on the tail probabilities of the form $P(F(l) > q)$ (where $F$ is a cumulative distribution function (CDF) and $l$ a random variable), and also a novel tight upper bound on $Var(F(l))$, which may be of independent interest.

Let's first establish some notations. Recall $P(\mathbf{x}, y, g)$ is the data-generating distribution (Appendix A.1). For the majority group $g = 0$ and minority-group $g = 1$, we denote $P_0(.) = P(.|g = 0)$ and $P_1(.) = P(.|g = 1)$ the data distribution for the majority and minority group, and recall $\gamma_0 = P(G = 0)$ and $\gamma_1 = 1 - \gamma_0$ the prevalence of these two groups. Furthermore:

**Group-specific error and its distribution**. Given a loss function $L(.,.)$ of Bergman divergence family and a fixed, unfair model $f_y$ that violates the sufficiency criteria (Arjovsky et al., 2019; Shui et al., 2022), denote $l_0 = L(\tilde{y}_0, f_y(\mathbf{x}_0))$ and $l_1 = L(\tilde{y}_1, f_y(\mathbf{x}_1))$ the cross-validation generalization gaps for the majority and the minority examples $(\tilde{y}_0, x_0) \sim P_0$ and $(\tilde{y}_1, x_1) \sim P_1$, where $\tilde{y}_g$ represents the true label without label noise. Notice that $(l_0, l_1)$ are random variables due to the randomness in $(P_0, P_1)$. To this end, also denote $(\mu_0, \mu_1), (\sigma_0^2, \sigma_1^2)$ the means and variances of the group-specific losses $(l_0, l_1)$. Due to the $f_y$'s violation of the sufficiency criteria (i.e., $E_0(y|f_y(\mathbf{x}) = t) \neq E_1(y|f_y(\mathbf{x}) = t)$), we expect the model $f_y$'s cross-validation loss is systematically worse for the minority groups, i.e., there's a systematic difference in group-specific loss $\mu_1 - \mu_0 = d > 0$. Finally, denote $F(l_i) = P(l < l_i)$ the CDF for the distribution of the loss $l = L(y, f_y(\mathbf{x})), (y, \mathbf{x}) \sim P$, and $(F_0, F_1)$ the CDF for the distributions of $l_0, l_1$, respectively.

**Rank-based Estimator**. We identify the minority group examples using a rank-based estimator. Specifically, recall $F(l_i) = P(l < l_i)$ is the CDF of the population loss, then the rank-based estimator for subgroup detection is:

$$\hat{I}(g_i = 1) = I(F(l_i) > q). \tag{18}$$

That is, we include a training example $\mathbf{x}_i$ into the introspective training only if the population quantile of its generalization error is higher than $q$, which is a user-specific threshold controlling the precision and recall of the estimator's identification performance.

Then, the below theorem describes how the estimator performance $P(g_i = 1|F(l_i) > q)$ is related to the user-specified threshold $q$, and the characteristics of data distribution (i.e., the group prevalence $\gamma_0, \gamma_1$) as well as the classifier performance (in terms of the group-specific loss distribution $F_0, F_1$).

**Theorem H.1** (Tail-group detection performance of rank-based estimator). *For $(l_0, l_1)$ a pair of random variables for minority- and majority-group examples. Denote $d = E(l_1 - l_0) > 0$ and $\sigma^2 = Var(l_1 - l_0)$ the mean and variance of the between-group generalization gap. Then, given a user-specified threshold $q \in (0, \frac{d^2}{d^2 + \sigma^2})$, the performance of the rank-based estimator $P(g = 1|F(l) > q)$ is bounded by:*

$$P(g = 1|F(l) > q) \geq (1 - \gamma_0)^2 + \gamma_0 * \frac{1 - \gamma_0}{1 - q} * \frac{z^2}{z^2 + 1} \quad where \quad z = \frac{E[F_0(l_1)] - q}{\sqrt{Var[F_0(l_1)]}}. \tag{19}$$

*Here, $F_0(l_1) = P(l_0 < l_1)$ is the majority-group loss CDF $F_0$ evaluated at the minority-group loss $l_1 = L(\tilde{y}_1, f_y(\mathbf{x}_1))$ where $(\tilde{y}_1, \mathbf{x}_1) \sim P_1$.*

Proof is at Appendix J. Notice here due to the randomness in $l_1$, $F_0(l_1)$ is a random variable that follows a continuous distribution and is bounded within $F_0 \in [0, 1]$. Therefore $F_0(l_1)$ has valid moments $E[F_0(l_1)]$ and $Var[F_0(l_1)]$. We see that the performance bound (19) is intuitively sensible: a lower majority group prevalence $\gamma_0$, a higher rank threshold $q$, and a higher likelihood for loss dominance $F_0(l_1) = P(l_0 < l_1)$ all contributes a stronger identification performance. Here, we see that $z = \frac{E[F_0(l_1)] - q}{\sqrt{Var[F_0(l_1)]}}$ is a distribution dependent quantity that governs the difficulty of identifying the minority group.

Clearly, a larger magnitude of $z$ leads to a stronger guarantee in detection performance in Theorem H.1. For the interested readers, the below result provides a lower bound for the magnitude of $z$ in terms of the characteristics of model performance (i.e., the moments of the group-specific loss distributions

(e.g., $d = \mu_1 - \mu_2$ and $(\sigma_1^2, \sigma_2^2)$), which can be used to obtain a more precise understanding of $P(g = 1 | F(l) > q)$ in (Equation (19)) in practice:

**Theorem H.2** (Lower bound on the standardized likelihood of loss dominance $z$.)**.** *Consider the standardized likelihood of loss dominance $z = \frac{E[F_0(l_1)] - q}{\sqrt{Var[F_0(l_1)]}}$, where the likelihood of loss dominance $F_0(l_1) = P(l_0 < l_1)$ is a random variable in terms of $l_1 = L(\tilde{y}_1, f_y(\mathbf{x}_1)), (\tilde{y}_1, \mathbf{x}_1) \sim P_1$. We have:*

$$E[F_0(l_1)] \geq \frac{d^2}{\sigma^2 + d^2} \quad where \quad d = \mu_1 - \mu_0 > 0, \ \sigma^2 = Var(l_1 - l_2) \leq \sigma_0^2 + \sigma_1^2,$$

$$Var[F_0(l_1)] \leq \frac{F_0(\mu_0)^2}{4} * \frac{\sigma_1^2}{\sigma_1^2 + d^2}, \tag{20}$$

*where $F_0(\mu_0) = P(l_0 \leq \mu_0)$ is the probability of the majority-group loss $l_0$ smaller than its mean $\mu_0 = E(l_0)$. Therefore, $z$ can be bounded by:*

$$z = \frac{E[F_0(l_1)] - q}{\sqrt{Var[F_0(l_1)]}} \geq \frac{2}{F_0(\mu_0)} * \sqrt{\frac{\sigma_1^2 + d^2}{\sigma_1^2}} * \left( \frac{d^2}{(\sigma_1^2 + \sigma_2^2) + d^2} - q \right). \tag{21}$$

The proof is in Appendix J. It relies on a novel upper bound of CDF variance (i.e., Equation (20)), which is important for guaranteeing a high magnitude of $z$ and consequently a tighter bound for group detection performance in Equation (19).

The combination of Theorems H.1-H.2 provides us an opportunity to quantitatively understand of the group detection performance $P(g = 1 | F(l) > q)$ in terms of the estimator configuration (i.e., the user-specified threshold $q$), data distribution (i.e., the majority-group prevalence $\gamma_0$), and the degree of unfairness of the vanilla model $f_y$ (in terms of the distributions of the group-specific losses). For example, consider a setting with majority-group prevalence $\gamma_0 = 0.85$, expected between-group loss gap $d = 1$, group-specific variances $\sigma_1 = \sigma_2 = 0.15$ and $P(l_0 < \mu_0) = 0.5$, using a percentile threshold $q = 0.9$, the rank-based estimator $I(F(l) > q)$ has group detection performance $P(g = 1 | F(l) > q) > 0.9$.

It is also worth commenting that, following the previous discussion, we see that the success of group identification relies on valid estimation of model's generalization error (with respect to the true label). To this end, the K-fold cross-validated ensemble procedure we used in this work is known to produce unbiased and low-variance estimates for the expected generalization loss (Blum et al., 1999; Kumar et al., 2013).

Finally, we highlight that the above results are broadly applicable and derived under weak, nearly assumption-free conditions. A even tighter performance bounds can be obtained by making further assumptions on the family of data distributions or model class, which is outside the scope of the current work. Another interesting direction is to incorporate the finite-sample estimation error of K-fold cross validation into the analysis (Blum et al., 1999; Kumar et al., 2013; Bayle et al., 2020), although this necessitates a careful treatment of model's generalization behavior (i.e., loss stability) in the subgroup setting, which we will pursue in the future work.

## I PROOF OF THEOREM 1

*Proof.* Let $r(\alpha) = (\alpha n)^{-p}$ and $r_{\text{agg}}(n) = \tau n^{-q} + \delta$. As discussed in section 3, we assume that the group-specific risk decays as

$$E[R(\hat{f}_{\alpha,n} | G = g)] = c_g r(\alpha_g) + r_{\text{agg}}(n)$$

where $\hat{f}_{\alpha,n}$ is the minimizer of the $\gamma$-weighted empirical risk on a dataset of size $n$ with allocation $\alpha$ and the expectation is with respect to the randomness of the dataset used to train $\hat{f}$.

Under the above assumption, the optimization problem for $\alpha$ looks like

$$\min_{\alpha \in \Delta^{|\mathcal{G}|}} \omega \left( \sum_g \gamma_g \left( c_g r(\alpha_g) + r_{\text{agg}}(n) \right) \right) + (1 - \omega) \max_g \left( c_g r(\alpha_g) + r_{\text{agg}}(n) \right)$$

The term $r_{\text{agg}}(n)$ is a constant that does not impact the optimal solution. Hence, we drop it and focus on the problem

$$\min_{\alpha \in \Delta^{|\mathscr{G}|}} \omega \left( \sum_g \gamma_g \left( c_g r(\alpha_g) \right) \right) + (1 - \omega) \max_g \left( c_g r(\alpha_g) \right)$$

We begin by noting that the objective is strictly convex and the domain of optimization is bounded. Hence, there exists a unique global optimum.

Turning the constraint $\sum_{g \in \mathscr{G}} \alpha_g = 1$ into a Lagrangian, we obtain

$$\min_{\alpha \geq 0} \omega \left( \sum_g \gamma_g \left( c_g r(\alpha_g) \right) \right) + (1 - \omega) \max_g \left( c_g r(\alpha_g) \right) + \lambda \left( \sum_{g \in \mathscr{G}} \alpha_g - 1 \right)$$

where $\lambda \in \mathbb{R}$ is a Lagrange multiplier.

At the optimum, we know that the Lagrangian should have 0 within its subdifferential wrt $\alpha$ (Boyd & Vandenberghe, 2004). The subdifferential is given by

$$\left\{ \begin{pmatrix} (\omega\gamma_1 + (1-\omega)\mu_1) c_1 r'(\alpha_1) + \lambda \\ (\omega\gamma_2 + (1-\omega)\mu_2) c_2 r'(\alpha_2) + \lambda \\ \vdots \\ (\omega\gamma_{|\mathscr{G}|} + (1-\omega)\mu_{|\mathscr{G}|}) c_{|\mathscr{G}|} r'(\alpha_{|\mathscr{G}|}) + \lambda \end{pmatrix} \text{ where } \mu \in \Delta^{|\mathscr{G}|} \text{ is such that } \mu_g > 0 \iff g \in \arg\max_{g' \in \mathscr{G}} c_{g'} r(\alpha_{g'}) \right\}$$

where $r'$ denotes the derivative of $r$. Since 0 belongs to the subdifferential at the optimum, there must exist $\mu$ satisfying the constraints above such that

$$\alpha_g^\star = r_{\text{inv}}' \left( -\frac{\lambda}{c_g \left( \omega\gamma_g + (1-\omega)\mu_g \right)} \right) \quad \forall g \in \mathscr{G}$$

where $r_{\text{inv}}'$ is the inverse of $r'$. Since $r'$ is a homogeneuous function of its argument, so is its inverse, and $\lambda$ can be eliminated to enforce the constraint $\sum_g \alpha_g = 1$, so that the optimal solution $\alpha_g^\star$ is

$$\alpha_g^\star = \frac{r_{\text{inv}}' \left( -\left( c_g \left( \omega\gamma_g + (1-\omega)\mu_g \right) \right)^{-1} \right)}{\sum_{g' \in \mathscr{G}} r_{\text{inv}}' \left( -\left( c_{g'} \left( \omega\gamma_{g'} + (1-\omega)\mu_{g'} \right) \right) \right)}$$

Let $s$ denote the denominator and $\theta_g = \omega + (1-\omega)\frac{\mu_g}{\gamma_g}$ so that

$$\alpha_g^\star = \frac{r_{\text{inv}}' \left( -\left( c_g \gamma_g \theta_g \right)^{-1} \right)}{s}$$

and $\theta$ must satisfy:

$$\sum_{g \in \mathscr{G}} \theta_g \gamma_g = 1, \theta_g \geq \omega \quad \forall g \in \mathscr{G}, \theta_g > \omega \iff g \in \arg\max_{g' \in \mathscr{G}} c_{g'} r\left( \alpha_{g'}^\star \right)$$

Using the fact that $r(t) = (nt)^{-p}$, we have that

$$\arg\max_{g'} c_{g'} r\left( \alpha_{g'}^\star \right) = \arg\min_{g'} c_{g'}^{-\frac{1}{p}} \alpha_{g'}^\star = \arg\min_{g'} c_{g'}^{\frac{1}{p+1} - \frac{1}{p}} \left( \gamma_g \theta_g \right)^{\frac{1}{p+1}} = \arg\min_{g'} c_{g'}^{-\frac{1}{p}} \gamma_g \theta_g$$

If we sort groups in ascending order according to $c_g^{-\frac{1}{p}} \gamma_g$ to obtain $g_1, g_2, \ldots$, a value of $\theta$ satisfying the conditions above can be computed as follows:
Initialize $\theta_g = \omega \quad \forall g \in \mathscr{G}$
Set $k = 1$
Until $\sum_g \theta_g \gamma_g = 1$ or $k = |\mathscr{G}|$ repeat:

a) $l = \min(k+1, |\mathscr{G}|)$

b) Set $\theta_{g_j} = \theta_{g_j} t$ (for $j \le k$) for the largest $t \ge 1$ such that (i) $\sum_g \theta_g \gamma_g = 1$ or (ii) $c_{g_1}^{-\frac{1}{p}} \gamma_{g_1} \theta_{g_1} = \ldots = c_{g_l}^{-\frac{1}{p}} \gamma_{g_l} \theta_{g_l}$

c) $k = k+1$

It is easy to see that this algorithm must terminate as it can go for at most $|\mathscr{G}|$ rounds. Further, by construction, we have $\theta_g \ge \omega \forall g \in \mathscr{G}$ since we start at these values and only scale up any of the $\theta_g$.

At the $j$-th iteration of the loop, we have that

$$c_{g_1}^{-\frac{1}{p}} \gamma_{g_1} \theta_{g_1} = \ldots = c_{g_j}^{-\frac{1}{p}} \gamma_{g_j} \theta_{g_j}$$

In the $j$-th iteration, we scale all $\theta_{g_j}$ ($m = 1, \ldots, j+1$) up by the same factor $t$ by the same amount so that we achieve the above for all groups upto $g_{j+1}$, or we hit the constraint $\sum_g \theta_g \gamma_g = 1$. If the former happens, we begin the next iteration with the same invariant. If the latter happens, we have obtained a $\theta$ that satisfies $\sum_g \theta_g \gamma_g = 1$ and $\theta_{g_i} > \omega$ for $i < k$ and $g_i \in \arg\min_{g'} c_{g'}^{-\frac{1}{p}} \gamma_g \theta_g$ for $i < k$.

If we reach the iteration where $k = |\mathscr{G}|$, we can simply scale up all the $\theta_g$ by the same amount until $\sum_g \theta_g \gamma_g = 1$ is satisfied and we would have that all $c_g^{-\frac{1}{p}} \gamma_g \theta_g$ is equal for all $g \in \mathscr{G}$ and all $\theta_g > \omega$.

$\square$

## J  PROOF OF THEOREMS H.1 AND H.2

### J.1  PROOF FOR THEOREM H.1

*Proof.* By Bayes' rule, we have:

$$P(g = 1 | F(l) > q) = P(F(l) > q | g = 1) * \frac{P(g = 1)}{P(F(l) > q)} \tag{22}$$

Expression for $P(g = 1)$ and $P(F(l) > q)$

Notice that $P(g = 1) = 1 - \gamma_0$, and $P(F(l) > q) = P(U > q) = 1 - q$ for $U \sim Unif(0,1)$. Here we used the fact that $F(l) \sim Unif(0,1)$ when $F$ is the CDF for the distribution of the random variable $l$ (Blitzstein & Hwang, 2015).

Expression for $P(F(l) > q | g = 1)$

Notice $P(F(l) > q | g = 1) = P(F(l_1) > q)$, where $l_1 = L(\tilde{y}_1, f_y(\mathbf{x}_1)), (\tilde{y}_1, \mathbf{x}_1) \sim P_1$ is the random variable for the minority-group loss. Also notice that $F(l_1) = P(l' < l_1)$ where $l' = L(y', f_y(\mathbf{x}'))$ for $(y', \mathbf{x}', g') \sim P$ is the non-group-specific loss. Then a conditional decomposition of $P(F(l_1) > q)$ reveals :

$$\begin{aligned}
P(F(l_1) > q) &= P[P(l' < l_1) > q] \\
&= P[P(l' < l_1) > q | g' = 0] P(g' = 0) + P[P(l' < l_1) > q | g' = 1] P(g' = 1) \\
&= P(F_0(l_1) > q) \gamma_0 + (1-q)(1-\gamma_0),
\end{aligned}$$

where the last equality follows since $P(g' = 0) = \gamma_0, P(g' = 1) = 1 - \gamma_0$ and $P[P(l' < l_1) > q] = P[F_1(l_1) > q] = P(U > q) = 1 - q$ for $U \sim Unif(0,1)$. Here we again used the fact that $F_1(l_1) \sim Unif(0,1)$ when $F_1$ is the CDF for the distribution of $l_1$ (Blitzstein & Hwang, 2015).

Lower bound for $P(F_0(l_1) > q)$

Lower bound $P(F_0(l_1) > q)$ by deriving a Cantelli-type inequality using the second moment method (Lyons & Peres, 2017). Specifically, for a random variable $R$ with $E(R) \ge 0$, we have:

$$E(R)^2 \le E(R * 1_{R>0})^2 \le E(R^2)P(R > 0) \quad \text{which implies} \quad \frac{P(R > 0)}{1 - P(R > 0)} \ge \frac{E(R)^2}{Var(R)}.$$

Setting $R = F_0(l_1) - q$, we have:

$$\frac{P(F_0(l_1) > q)}{1 - P(F_0(l_1) > q)} \geq \frac{(E(F_0(l_1)) - q)^2}{Var(F_0(l_1))} = z^2, \quad \text{which implies} \quad P(F_0(l_1) > q) \geq \frac{z^2}{1 + z^2}$$

### Derive the final bound

Finally, using the above three facts, we can express Equation (22) as:

$$P(g = 1 | F(l) > q) = P(F(l) > q | g = 1) * \frac{P(g = 1)}{P(F(l_1) > q)} = P(F(l) > q | g = 1) * \frac{1 - \gamma_0}{1 - q}$$

which further leads to:

$$P(g = 1 | F(l) > q) = \left( (1 - q)(1 - \gamma_0) + P(F_0(l_1) > q)\gamma_0 \right) * \frac{1 - \gamma_0}{1 - q} \geq \left( (1 - q)(1 - \gamma_0) + \frac{z^2}{1 + z^2} * \gamma_0 \right) * \frac{1 - \gamma_0}{1 - q},$$

yielding the final bound $P(g = 1 | F(l) > q) \geq (1 - \gamma_0)^2 + \gamma_0 * \frac{1 - \gamma_0}{1 - q} * \frac{z^2}{1 + z^2}$ as in Equation (19). $\qquad \square$

## J.2 PROOF FOR THEOREM H.2

*Proof.* For $F_0(l_1)$, recall $F_0$ is the CDF of $l_0 = L(\tilde{y}_0, f_y(\mathbf{x}_0))$ where $(\tilde{y}_0, \mathbf{x}_0) \sim P_0$ and $l_1 = L(\tilde{y}_1, f_y(\mathbf{x}_1))$ is the random variable of minority-group loss with $(\tilde{y}_1, \mathbf{x}_1) \sim P_1$.

To derive lower bound for $z = \frac{E(F_0(l_1)) - q}{Var(F_0(l_1))}$, first derive bounds on $E(F_0(l_1))$ and $Var(F_0(l_1))$:

### Lower bound for $E(F_0(l_1))$

Recall $P(F(l) | g = 1) = P(F(l_1))$, where $l_1 = L(\tilde{y}_1, f_y(\mathbf{x}_1))$ is the random variable for minority-group error with $(\tilde{y}_1, f_y(\mathbf{x}_1)) \sim P_1$. Then:

$$E(F_0(l_1)) = E(F_0(l) | g = 1) = P(l_0 \leq l_1) = P(l_1 - l_0 \geq 0).$$

By the second moment method inequality (Lyons & Peres, 2017), we have:

$$\begin{aligned} P(l_1 - l_0 \geq 0) &\geq \frac{E[l_1 - l_0]^2}{E[(l_1 - l_0)^2]} = \frac{E[l_1 - l_0]^2}{Var(l_1 - l_0) + E[l_1 - l_0]^2} \\ &\geq \frac{E[l_1 - l_0]^2}{Var(l_1) + Var(l_0) + E[l_1 - l_0]^2} \\ &= \frac{d^2}{(\sigma_0^2 + \sigma_1^2) + d^2}. \end{aligned} \qquad (23)$$

### Upper bound for $Var(F_0(l_1))$

Recall that $Var(F(l)) = E\left[ (F(l) - E(F(l)))^2 \right]$. Using the iterative expectation formula, we split the variance computation for $F_0(l_1)$ into two regions of $l_1$ depending on whether the minority-group loss of $l_1 \geq \mu_1 - \lambda \sigma_1$:

$$\begin{aligned} Var(F_0(l_1)) = &Var(F_0(l_1) | l_1 \leq \mu_1 - \lambda \sigma_1)P(l_1 \leq \mu_1 - \lambda \sigma_1) + \\ &Var(F_0(l_1) | l_1 > \mu_1 - \lambda \sigma_1)P(l_1 > \mu_1 - \lambda \sigma_1), \end{aligned}$$

which holds for any positive multiplier $\lambda > 0$.

Notice that in the above, $P(l_1 < \mu_1 - \lambda \sigma_1)$ in the second line describes the tail probability of the loss distribution of $l_1$, for suitably large $\lambda$, $P(l_1 < \mu_1 - \lambda \sigma_1)$ should be small. On the other hand, $Var(F_0(l_1) | l_1 \geq \mu_1 - \lambda \sigma_1)$ in the second line describes the variance of $F_0(l_1)$ in the region where $l_1$ is large. When the distribution of $l_0$ and $l_1$ is well separated (i.e., $\mu_1 - \mu_0 = d > 0$), for suitable value of $\lambda$, we expect the value of $F_0(l_1 = F_0(\mu_1 - \lambda \sigma_1))$ to be high and close to 1, and as a result the $Var(F_0(l_1) | l_1 \geq \mu_1 - \lambda \sigma_1)$ will be low since $F_0(l_1)$ is bounded within a small range $[F_0(\mu_1 - \lambda \sigma_1), 1]$.

Consequently, to obtain a tight upper bound of $Var(F(l))$, it is sufficient to identify a suitable value of $\lambda$ such that both $Var(F_0(l_1)|l_1 \geq \mu_1 - \lambda\sigma_1)$ and $P(l_1 < \mu_1 - \lambda\sigma_1)$ are low.

To identify a suitable value of $\lambda$, first derive an upper bound of $Var(F_0(l_1))$ in terms of $\lambda$. Notice below two facts:

- By Cantelli's inequality:
$$P(l_1 \leq \mu_1 - \lambda\sigma_1) \leq \frac{1}{\lambda^2 + 1}.$$

- By the inequality for variance of the bounded variables:
$$Var(F_0(l_1)|l_1 \leq \mu_1 - \lambda\sigma_1) \leq \frac{1}{4}F_0(\mu_1 - \lambda\sigma_1)^2,$$

  where the first inequality follows by the fact that for a random variable $R$ bounded between $[a,b]$ (in this case between $[F(\mu_1 - \lambda\sigma_1), 1]$), its variance is bounded by $Var(R) \leq (b - E(R))(E(R) - a) \leq \frac{1}{4}(b-a)^2$.

- By Markov's inequality, $F_0(l_1) = P(l_1 \leq l_1) \geq \mu_0/l_1$, which implies:
$$Var(F_0(l_1)|l_1 \geq \mu_1 - \lambda\sigma_1) \leq \frac{1}{4}(1 - F_0(\mu_1 - \lambda\sigma_1))^2 \leq \frac{1}{4}(1 - \frac{\mu_0}{\mu_1 - \lambda\sigma_1})^2,$$

  where the first inequality also follows by the variance inequality of the bounded variables.

Using the above two facts, we can bound $Var(F_0(l_1))$ as:

$$Var(F_0(l_1)) = Var(F_0(l_1)|l_1 \leq \mu_1 - \lambda\sigma_1) * P(l_1 \leq \mu_1 - \lambda\sigma_1) + Var(F_0(l_1)|l_1 > \mu_1 - \lambda\sigma_1) * P(l_1 \geq \mu_1 - \lambda\sigma_1)$$

$$\leq Var(F_0(l_1)|l_1 \leq \mu_1 - \lambda\sigma_1) * \frac{1}{\lambda^2 + 1} + Var(F_0(l_1)|l_1 > \mu_1 - \lambda\sigma_1) * \frac{\lambda^2}{\lambda^2 + 1}$$

$$\leq \frac{1}{4}F_0(\mu_1 - \lambda\sigma_1)^2 * \frac{1}{\lambda^2 + 1} + \frac{1}{4}(1 - \frac{\mu_0}{\mu_1 - \lambda\sigma_1})^2 * \frac{\lambda^2}{\lambda^2 + 1}. \tag{24}$$

where the first inequality holds since the first conditional variance term $Var(F_0(l_1)|l_1 \leq \mu_1 - \lambda\sigma_1)$ (i.e., variance in the majority bulk) is expected to be much larger than the second term $Var(F_0(l_1)|l_1 > \mu_1 - \lambda\sigma_1)$ (i.e., variance in the far tail), therefore assigning the highest possible probability weight $P(l_1 \leq \mu_1 - \lambda\sigma_1)$ to the larger variance term $Var(F_0(l_1)|l_1 \leq \mu_1 - \lambda\sigma_1)$ leads to an upper bound for $Var(F_0(l_1))$.

Further simplifying Equation (24), we arrive at:

$$Var(F_0(l_1)) \leq \frac{1}{4}F_0(\mu_1 - \lambda\sigma_1)^2 * \frac{1}{\lambda^2 + 1} + \frac{1}{4}(1 - \frac{\mu_0}{\mu_1 - \lambda\sigma_1})^2 * \frac{\lambda^2}{\lambda^2 + 1}$$

$$= \frac{1}{4} * \frac{1}{\lambda^2 + 1} * \left(F_0(\mu_1 - \lambda\sigma_1)^2 + \frac{\lambda^2}{4}(1 - \frac{\mu_0}{\mu_1 - \lambda\sigma_1})^2\right)$$

$$= \frac{1}{4} * \frac{1}{\lambda^2 + 1} * \left(F_0(\mu_1 - \lambda\sigma_1)^2 + \frac{\lambda^2}{4}(\frac{d - \lambda\sigma_1}{\mu_1 - \lambda\sigma_1})^2\right)$$

where recall $d = \mu_1 - \mu_0$ is the expected error gap between the majority and the minority groups. Setting $\lambda = \frac{d}{\sigma_1}$, we have:

$$Var(F_0(l_1)) \leq \frac{1}{4} * F_0(\mu_0)^2 * \frac{1}{\lambda^2 + 1} = \frac{F_0(\mu_0)^2}{4} * \frac{\sigma_1^2}{d^2 + \sigma_1^2}. \tag{25}$$

Lower bound for $z$

Finally, plugging the bounds for $E(F_0(l_1))$ and $Var(F_0(l_1))$ from Equation (23) and 25 into $z = \frac{E[F_0(l_1)]-q}{\sqrt{Var[F_0(l_1)]}}$ yields the final lower bound in Equation (21), i.e.,

$$z = \frac{E[F_0(l_1)]-q}{\sqrt{Var[F_0(l_1)]}} \geq \left(\frac{d^2}{(\sigma_1^2+\sigma_2^2)+d^2} - q\right) \Big/ \sqrt{\frac{F_0(\mu_0)^2}{4} * \frac{\sigma_1^2}{d^2+\sigma_1^2}}$$

$$= \frac{2}{F_0(\mu_0)} * \sqrt{\frac{\sigma_1^2+d^2}{\sigma_1^2}} * \left(\frac{d^2}{(\sigma_1^2+\sigma_2^2)+d^2} - q\right).$$

$\square$

