# OpenReview forum: "Pushing the Accuracy-Group Robustness Frontier with Introspective Self-play"
_ICLR.cc/2023/Conference — ICLR 2023 poster_

### Official Review · Reviewer_dJid · 2022-10-22

**Confidence:** 2
**Correctness:** 4
**Technical Novelty And Significance:** 3
**Empirical Novelty And Significance:** 3
**Recommendation:** 8

**Clarity, Quality, Novelty And Reproducibility:**

I think this was a clearly-written, well-motivated, nicely-executed paper. The selected baselines for comparison against were reasonable, and the theoretical results were convincing to me.

Some questions for the authors:
- It would still be interesting to see results on DRO given that DRO is still a very popular method.
- What happens when batch sizes are very small, say the batch size is 24 and the underrepresented group is only 1% of the data, so the batch only has 2 or 3 samples?
- Have the authors tried out the training objective with other losses besides CE? Apologies if I missed this in the paper.

**Strength And Weaknesses:**

Strengths:
- Interesting idea of adding another task to predict if a sample is underrepresented to tackle the accuracy-fairness tradeoff problem
- Empirical evaluation included state-of-the-art baselines (e.g. RWT, JTT) and the method outperformed them.

Weaknesses:
- Results should be accompanied by standard deviations so that we can judge if the results are significantly different



**Summary Of The Paper:**

The paper introduces a method called introspective self-play that uses multitask techniques (add an auxiliary task that predicts bias for each data point) to make the resulting model more accurate and fair.

**Summary Of The Review:**

I found the paper very interesting and well-executed. I think it could be improved even further with some tweaks, like adding confidence intervals, and some investigation of what happens when batch sizes are very, very small.

---

> ### Author Response · Authors · 2022-11-17
> **Response to reviewer dJid**
>
> Thanks for your encouraging comments! We are grateful to see [dJid](https://openreview.net/forum?id=MofT9KEF0kw&noteId=xpGGqlHHxTS) recognizing our theoretical contribution and empirical evaluation, and find the paper well-motivated, nicely-executed. We also appreciate the additional suggestions you have made, which we respond below:
>
> * > It would still be interesting to see results on DRO given that DRO is still a very popular method.
>
> Thanks! Following your suggestion, we have implemented the DRO baseline for the Census Income and Toxicity Detection experiments, and added them to Table 2. As shown there, the DRO-trained active learning model underperforms ISP-Identity in acquiring the tail-group examples (tail-group sampling rate 0.755 (DRO) v.s 0.907 (ISP-Identity) for Census Income, and 0.841 (DRO) v.s 0.905 (ISP-Identity) for Toxic Comments).
>
> * > Results should be accompanied by standard deviations.
>
> Thanks, following your suggestion, we are currently conducting 10 replica runs of all active learning experiments to compute the standard deviation in tail sampling rate and also final performance. The experiments for Census Income is completed and the result are as below (standard deviation in brackets):
>
> | Method | Tail Sampling Rate | Combined Acc | Worst Group Acc
> | --- | :---: | :---: | :---: |
> | **(Random)** | 0.475 (0.005) | 0.746 (0.020) | 0.659 (0.018) |
> | **RWT** | 0.797 (0.010) | 0.772 (0.015) | 0.761 (0.020) |
> | **DRO** | 0.755 (0.009) | 0.759 (0.012) | 0.729 (0.018) |
> | **ISP-Identity (Ours)** | 0.907 (0.011) | 0.785 (0.008) | 0.774 (0.021) |
> | ------------ | ------------ | ------------ | ------------ |
> | **ERM** | 0.791 (0.011) | 0.736 (0.012) | 0.658 (0.022) |
> | **JTT** | 0.839 (0.012) | 0.752 (0.012) | 0.695 (0.025)|
> | **ISP-Gap (Ours)** | 0.839 (0.012) | 0.770 (0.011) | 0.753 (0.023) |
>
> The Toxic Comments experiments are still running, and we will add the standard deviation result to the paper when preparing the camera ready.
>
> * > Have the authors tried out the training objective with other losses besides CE? Apologies if I missed this in the paper.
>
> Thanks for this question. In this initial investigation, we have focused on the canonical setting in the group robustness literature (i.e., classification under CE loss). However, it is indeed interesting to investigate other tasks (e.g., regression under squared loss) or advanced classification losses (e.g., focal loss or generalized cross entropy).
>
> To incorporate this point, we have highlighted investigating additional loss functions as future work in the conclusion section (Section 5) of the updated manuscript.
>
> * > What happens when batch sizes are very small, say the batch size is 24 and the underrepresented group is only 1% of the data, so the batch only has 2 or 3 samples?
>
> Thanks for this question. Due to the focus of large-scale setting in this current work, we have prioritized the large-scale, large-batch active learning setting in the experiment evaluation, which is closer to the type of real-world problems we intend to make progress on.
>
> However, although the small-batch setting is out of scope of this initial work, we agree that it is interesting to consider the possible impact of batch size on the quality of the introspective training. For example, it is likely that setting the batch size too small may make the under-representation head unstable during the early stage of introspective training, and therefore impacting the model performance.
>
> To incorporate this point, we have included this as a future direction in the conclusion section (Section 5) of the updated manuscript, and will leave a full numerical study of model performance with respect to different training details (e.g., batch size) to a journal version of the paper.

---

### Official Review · Reviewer_zGVn · 2022-10-23

**Confidence:** 4
**Correctness:** 3
**Technical Novelty And Significance:** 3
**Empirical Novelty And Significance:** 3
**Recommendation:** 6

**Clarity, Quality, Novelty And Reproducibility:**

Clarity

This paper is well organized.

Quality

The paper is technically solid.

Novelty

The novelty of this paper is reasonable.

Reproducibility

Good.

**Strength And Weaknesses:**

Strength

+The research of fairness is critical to the field of deep learning.

+ Experiment results show reasonable improvement.

Weaknesses

- On page 2, "Our key observation is that given a sampling model with well-calibrated uncertainty (i.e., the model uncertainty is
well-correlated with generalization error), active learning (AL) can preferentially acquire tail-group examples from unlabelled data without needing group annotations, and add them to the training data to reach a more balanced data distribution" needs to be discussed with more details, especially its relationship to fairness.

- Please include a color legend in Figure 2, 6, 7

- In Table 2, if combined accuracy is defined as (accuracy + worst-group accuracy)/2, why is worst-group accuracy higher than combined accuracy and accuracy (?)?

-In Table 2, ISP-Gap achieves almost identical Tail Sampling Rate as JTT, why does it perform better? More discussion and details are needed.

**Summary Of The Paper:**

To Improve the accuracy-fairness frontier of DNN models, This paper presents a technique called Introspective Self-play, which estimates
DNN uncertainty by adding an auxiliary introspection task requiring DNN to predict the bias for each data point in addition to the label.

**Summary Of The Review:**

This paper is technically solid, and its novelty is reasonable.

---

> ### Author Response · Authors · 2022-11-16
> **Response to Reviewer zGVn**
>
> Thanks for your insightful comments! We have updated paper with additional discussion and correction thanks to your comment.
>
> Detailed response below:
>
> 1. Discuss relationship to fairness
>
> > On page 2, "Our key observation is that given a sampling model with well-calibrated uncertainty (i.e., the model uncertainty is well-correlated with generalization error), active learning (AL) can preferentially acquire tail-group examples from unlabelled data without needing group annotations, and add them to the training data to reach a more balanced data distribution" needs to be discussed with more details, especially its relationship to fairness.
>
> Thanks for this comment. We have added the following discussion to Appendix A.5. to stress the connection to fairness, and point to it in the introduction:
>
> ``In the context of fairness-aware learning, this means a sampling model with calibrated uncertainty preferentially samples the under-represented minority group, leading to a sampled training set with more balanced representation among population subgroups, and consequently reducing the between-group generalization gap in the trained model,promoting its fairness properties."
>
> As part of response to other reviewers, we also included additional discussion about the subgroup performance vs fairness terminology in the literature, which we have also included in Appendix A.5. and included below:
>
> "The fairness definition we considered in this work (i.e., worst-group performance) corresponds to the notion of minimax fairness or Rawlsian max-min fairness ([Lahoti et al 2020](https://proceedings.neurips.cc/paper/2020/hash/07fc15c9d169ee48573edd749d25945d-Abstract.html), [Martinez et al 2020](https://proceedings.mlr.press/v119/martinez20a.html), [Martinez et al 2021](https://proceedings.mlr.press/v139/martinez21a.html), [Diana et al 2021](https://dl.acm.org/doi/abs/10.1145/3461702.3462523), [Rawls 2001](https://www.hup.harvard.edu/catalog.php?isbn=9780674005112), [Rawls 2003](https://www.taylorfrancis.com/chapters/edit/10.4324/9780203495667-33/theory-justice-john-rawls)). There have been notable works developed for both analyzing accuracy-fairness tradeoff under this notion (e.g., [Martinez et al 2020](https://proceedings.mlr.press/v119/martinez20a.html), [Martinez et al 2021](https://proceedings.mlr.press/v139/martinez21a.html)), and for improving fairness performance without explicit annotation (i.e., ARL, [Lahoti et al 2020](https://proceedings.neurips.cc/paper/2020/hash/07fc15c9d169ee48573edd749d25945d-Abstract.html)). It was also the original objective of IRM ([Arjovsky et al 2020](https://arxiv.org/pdf/1907.02893.pdf), Section 2)."
>
>
> 2. Clarification Questions:
>
> * > In Table 2, if combined accuracy is defined as (accuracy + worst-group accuracy)/2, why is worst-group accuracy higher than combined accuracy and accuracy?
>
> Thanks for highlighting this. This is caused by a logging error when transporting results to LaTeX (our apologies). In the revised manuscript, we have updated the numbers in Table 2 according to the correct number. As shown, there's no change to the relative trends between methods and our conclusion.
>
> * > In Table 2, ISP-Gap achieves almost identical Tail Sampling Rate as JTT, why does it perform better? More discussion and details are needed.
>
> Thanks for this question. The final model performance is a combination of two factors:  (1) the group distribution of the training data, and (2) the quality of the reweighting signal in the final training. Notice that as shown in Table 1, JTT and ISP-Gap are using different signals for final reweighting (i.e., training error v.s. generalization gap).
>
> As a result, although JTT and ISP-Gap achieve similar levels of distributional balance, ISP-Gap is able to attain stronger tradeoff performance in the final model by virtue of providing better reweighting signals for final-model reweighted training. In Section 4.2, we have also conducted an ablation study to further investigate the impact of signal quality to final model performance. As shown in Table 3, compared to using Error as reweighting signal (row 2), using Gap as reweighting signal consistently leads to stronger performance when combined with actively sampled (i.e., non-random) training datasets.

---

### Official Review · Reviewer_1jcC · 2022-10-25

**Confidence:** 5
**Correctness:** 3
**Technical Novelty And Significance:** 3
**Empirical Novelty And Significance:** 3
**Recommendation:** 8

**Clarity, Quality, Novelty And Reproducibility:**

In general this paper is clearly written and easy to follow. The reproducibility seems high.


**Strength And Weaknesses:**

### Pros

- In general, this paper proposed a quite meaningful solution in addressing the worst group performance: active learning by querying the additional minority group samples. Moreover, the scenario is much more difficult than the conventional setting since we did not observe the subgroup information.  Overall I would think the problem is well-motivated and quite meaningful in real-world practice.
- The paper in general is well-written and clearly illustrated. The related work part is quite comprehensive and detailed. (I really like them!)
- The theoretical analysis is done and improved results in both toy and real-world language dataset.

### Cons:
- [Related work] Despite the detailed discussions on related works, I would feel some parts are seemingly incorrect and not clear.
- [About the title/main contribution] I would think the fair-accuracy is NOT a proper name in the fairness literature. The fairness is defined as worst-case group performance, which is generally different from the conventional fair notations such as demographic parity, equalized odd or sufficiency. That means this paper did not address the inherent trade-off in traditional fairness accuracy trade-off.
- [About subgroup detection] Despite the theoretical analysis being done, there are still many theoretical/fundamental concerns related to the identifiability of subgroup estimation.
- [About the practical scalability] I have doubts on the scalability of the proposed approach, because the bootstrapping approach would be quite long in a large scale dataset.

### Detailed Comments on cons

1. [Related work] Despite comprehensive analysis, I would still have several remarks on the related work. Indeed this part is for the purpose of discussion.

(1) In appendix D.1-D.3 learning representation under data bias. I would say the discussion about IRM and EIIL are seemingly inaccurate. In fact, IRM ensures a representation such that for all subgroups P(Y|Z,g) being invariant.  IRM v1 is further proposed to approximate the original objective through gradient penalty. Further EIIL adopted IRM_v1 to detect the unobserved subgroups.

Back to the assumptions within the paper (footnote in page 2), the data generation across different groups should be the same, which seems a bit strong assumption for me. Indeed, in the feature level (e.g, pixels in images), the environment distribution P(Y|X,g) could be completely different (e.g, classifying cat in different backgrounds). However, a more reasonable solution is by assume the existence of a representation function $f(x)=z$ such that P(Y|z,g) are the same. Intuitively, this assumes the semantic(or high-level) information such as cat/dog being invariant.  This is also related to fairness w.r.t sufficiency (see IRM[1], EIIL[2] and recent paper [3] for detailed discussions). Therefore I would think the data generation assumption could be better refined.

(2) About learning fair representation D3. I should say related work here requires better refinement. Indeed, learning fair representation generally is based on DP/EO/sufficiency (such as EIIL [2] and Paper[1,3,4]). I would say the current discussion does not make sufficient sense in fair learning. Since fair representation learning depends on the predefined fair notions, it could not be simplified as adversarial learning, regularization, etc..

(3) About related work in active learning under distribution shift in D4. Well, as far as I know, in active learning literature, the concept of distribution has been well developed recently such as paper [5,6,7]. They essentially model active learning between labelled and unlabeled dataset as a distribution shift problem.

(4) Related work. D5 Deep uncertainty methods in active learning. I would think the proposed approach are essentially improve the **diversity** rather than uncertainty in the paper (see paper [5-7] for details). Essentially, the reason for the unfair predictions is due to the limited samples within some subgroup. Thus diversity aims to search these samples by making the empirical distribution similar to the ground truth distribution. I would think this is well-aligned with your motivations.

2. [About the title/main contribution] I would think the fair-accuracy is NOT a proper name in the fairness literature. In this paper, the fairness is defined as worst-case group performance, which is generally different from conventional fair notions such as **demographic parity, equalized odd or sufficiency**. That means this paper did not address the inherent trade-off in traditional fairness accuracy trade-off.

In contrast, this paper essentially studied the prediction variance of learning from multiple subgroups. If the accuracy/loss variance of all subgroups are zero, the fairness (within your paper) is achieved. However, this notion is completely **DIFFERENT** from EO/DP/sufficiency, where they are directly defined on the predictor behaviors and could induce an inherent trade-off. From this viewpoint, I do think the title is over-claiming and could be potentially misleading for the traditional fairness community. It should clearly reformulate the real contribution by considering the worst-case group performance (such as the related paper in Distribution Robust optimization..)

3. [About subgroup detection] Despite the theoretical analysis being done, there are still many theoretical/fundamental concerns related to the identifiability of subgroup estimation. From the observational data, when could we identify the subgroup? Please note this is the key point within your paper. If we could not identify the correct subgroup, the whole learning procedure is incorrect. From the paper, the assumption seems like
$P(Y|X,g)$ being invariant as the underlying assumption. While if the data generation distribution is identical, there is no subgroup, right? I agree the empirical counterpart could be slightly different but this is quite difficult to detect the correct version, right? From this perspective, I do think there are sort of issues in the identifiability.

If we consider a mild assumption such as paper [3] by assuming data distribution $P(y|X,g)$ being different and representation distribution $P(y|Z,g)$ being invariant. This could be a better assumption and it seems that we could identify the subgroup information.

Overall, I do think the paper requires clear theoretical assumptions and illustrates why the cross-validation based approach could identify the subgroup (I am not sure the proposed approach is also ad-hoc or not)


4. [About the practical scalability] I have doubts on the scalability of the proposed approach, because the bootstrapping approach would be quite long in a large scale dataset such as a language dataset. What is the time complexity of that? It seems that we need multiple fine-tunings?

Ref
[1] Environment Inference for Invariant Learning. Icml 2021

[2] Invariant risk minimization, 2019

[3] Fair Representation Learning through Implicit Path Alignment. ICML 2022

[4] Out of Distribution Generalization in Machine Learning. Martin Arjovsky. 2020

[5] Deep active learning: Unified and principled method for query and training. Aistat 2020

[6] Discrepancy-Based Active Learning for Domain Adaptation. ICLR 2022

[7] Low-Budget Active Learning via Wasserstein Distance: An Integer Programming Approach. ICLR 2022


**Summary Of The Paper:**

This paper studied the problem of improving accuracy (performance of average subgroups) and fairness (performance in worst case subgroups). Then this paper further proposed an active learning strategy by querying the worst-subgroup to improve the accuracy-fairness boundary. Besides, this paper assumes that  subgroup information is hidden and designs a novel ensemble based approach to detect/estimate the subgroup information. The corresponding theoretical understanding is further derived. Finally empirical results verify the proposed framework.

-----------------
### Post-Rebuttal

**I would appreciate author responses. My concerns about related work, fair notions, and assumptions have been addressed.
In light of these results, I support acceptance.**

P.s Both the overall and correctness scores have been updated.

**Summary Of The Review:**

This paper studied the problem of improving accuracy (performance of average subgroups) and fairness (performance in worst case subgroups).

In general, I would think this paper is solid in the presentation and related work.  There are still several important concerns within the paper (see cons in the review), I would recommend a borderline paper.

---

> ### Author Response · Authors · 2022-11-16
> **Response to reviewer**
>
> Thanks for your insightful comments! We have updated the paper's positioning, data assumptions, descriptions in the related work, and added new theorems to incorporate your suggestions (highlighted in blue in the updated manuscript). Please let us know any of these changes are not clear or can be further improved -  we are happy to work with you to continue improving the paper.
>
> Detailed response below.
>
> 1. Related work
>
> * > (D.1-3) learning representation under data bias. I would say the discussion about IRM and EIIL are seemingly inaccurate. In fact, IRM ensures a representation such that for all subgroups P(Y|Z,g) being invariant. IRM v1 is further proposed to approximate the original objective through gradient penalty. Further EIIL adopted IRM_v1 to detect the unobserved subgroups.
>
> We have updated our descriptions about IRM / EIIL in D.1-3 using the language suggested. Please let me know if the description can be further improved - happy to update them.
>
> * > (D.3 learning fair representation). I should say related work here requires better refinement. Indeed, learning fair representation generally is based on DP/EO/sufficiency (such as EIIL [2] and Paper[1,3,4]). I would say the current discussion does not make sufficient sense in fair learning. Since fair representation learning depends on the predefined fair notions, it could not be simplified as adversarial learning, regularization, etc.
>
> Thanks for this comment. Given how active this research area is, there are some differences in nomenclature in the field, here we have followed the distinctions laid out in the literature review by [Canton and Haas 2020](https://arxiv.org/pdf/2010.04053.pdf) Section 4 and Figure 4; [Horte et al 2022](https://arxiv.org/pdf/2207.07068.pdf) Section 4.2.
> However, we acknowledge that when introducing this line of work, the role of fairness criteria (DP/EO/sufficiency) and the goal of fair representation learning as preserving the invariant correlations (between the embedding and true label) should be better stressed. We have restated D.3 by leading the section with a discussion of the pre-fined fairness notions and the goal of learning invariant correlations.
>
> * > (D.4 active learning under distribution shift.) ... in active learning literature, the concept of distribution has been well developed recently such as paper [5,6,7].
>
> Thanks, we have included this line of work in Section D.4.
>
> 2. About the title/main contribution
>
> * > In this paper, the fairness is defined as worst-case group performance, which is generally different from conventional fair notions such as demographic parity, equalized odd or sufficiency. … It should clearly reformulate the real contribution by considering the worst-case group performance (such as the related paper in Distribution Robust optimization..)
>
> Thanks for raising this point.  The fairness definition we use (i.e., worst-group performance) corresponds to the notion of minimax fairness or Rawlsian max-min fairness ([Lahoti et al 2020](https://proceedings.neurips.cc/paper/2020/hash/07fc15c9d169ee48573edd749d25945d-Abstract.html), [Martinez et al 2020](https://proceedings.mlr.press/v119/martinez20a.html), [Martinez et al 2021](https://proceedings.mlr.press/v139/martinez21a.html), [Diana et al 2021](https://dl.acm.org/doi/abs/10.1145/3461702.3462523), [Rawls 2001](https://www.hup.harvard.edu/catalog.php?isbn=9780674005112), [Rawls 2003](https://www.taylorfrancis.com/chapters/edit/10.4324/9780203495667-33/theory-justice-john-rawls)). There have been notable works developed for both analyzing accuracy-fairness tradeoff under this notion (e.g., [Martinez et al 2020](https://proceedings.mlr.press/v119/martinez20a.html), [Martinez et al 2021](https://proceedings.mlr.press/v139/martinez21a.html)), and for improving fairness performance without explicit annotation (i.e., ARL, [Lahoti et al 2020](https://proceedings.neurips.cc/paper/2020/hash/07fc15c9d169ee48573edd749d25945d-Abstract.html)). It was also the original objective of IRM ([Arjovsky et al 2020](https://arxiv.org/pdf/1907.02893.pdf), Section 2).
>
> With that said, we do agree minimax fairness does not cover all common notions of fairness such as demographic parity, equalized odd or sufficiency. To better communicate our core contribution and avoid misleading readers, we will use "group robustness" instead of "fairness" in the title, abstract and the main text (similar to the naming convention used in the [Just Train Twice](https://arxiv.org/abs/2107.09044) paper), and add a discussion of its connection to the fairness literature in the Appendix A.5.

---

> ### Author Response · Authors · 2022-11-16
> **Response to reviewer (Part 2)**
>
> 3. Theoretical treatment of subgroup detection performance
>
> * > "If we could not identify the correct subgroup, the whole learning procedure is incorrect. "
>
> Thanks for highlighting this nuanced point that is worth further elaboration.
>
> In fact, it is not necessary to perfectly identify all the subgroup examples to improve the final model's tradeoff frontier.
> For example, when there exists hard-to-learn majority-group examples and also easy-to-learn minority-group examples (also known as "benign bias" examples in the debiasing literature  [Nam et al (2022)](https://proceedings.neurips.cc/paper/2020/file/eddc3427c5d77843c2253f1e799fe933-Paper.pdf)), the active sampling budget is better spent in sampling some of the hard-to-learn majority examples over the easy-to-learn minority examples, so that the final tradeoff frontier is meaningfully improved both in the directions of subgroup performance and of overall accuracy. To this end, including some hard-to-learn majority examples (detected by the loss-based procedure) into introspective training help us to achieve that.
>
> This is exactly the case in our toxicity detection experiments (Table 2). As shown, comparing between ISP-Identity v.s. ISP-Gap (trained on true group label v.s. cross-validation estimated label), the ISP-Gap attains a significantly stronger accuracy-group robustness performance while sampling fewer minority examples. This is likely caused by the cross-validation estimator's ability to capture challenging, non-identity-related examples instead of the easy-to-learn and identity-related examples, which results in more sampling efficiency in the minority group (i.e., the identity-related comments), and improved tradeoff frontier for the final model.
>
> We have added this discussion in Appendix B.2, and pointed to it in the main paper.
>
> Further, we would like to highlight that the Theorem 1 of our paper provides another theoretical justification for the notions of hard-to-learn majority-group v.s. easy-to-learn minority group examples. There, the theorem shows that in an ideal scenario where one were allowed to optimize the representation of each subgroup having knowledge of the group-specific learning difficulty (as modeled by the learning rate function parameterized by c_g), the optimal strategy is to systematically over-sample groups that have low normalized representation, i.e., groups for which the representation normalized by the learning difficulty is low. While this ideal scenario deviates from practice, it strongly motivates a strategy based on identifying groups with low normalized learning difficulty and systematically up-sampling them via active learning -ISP attempts to do exactly this.
>
> * > If we consider a mild assumption such as paper [3] by assuming data distribution P(y|X,g) being different and representation distribution P(y|Z,g)=P(y|Z) being invariant.
>
> Thanks for this suggestion. We agree that this mild assumption is more appropriate, and we have updated the condition $P(y|X,g)=P(y|X)$ to $P(y|Z,g)=P(y|Z)$ in the **Notation and Problem Setup** paragraphs of Section 1 and Appendix A.1.
> Fortunately, this change does not impact to the rest of the paper, since the current theoretical results (i.e., Proposition 1 and Theorem 1) do not require the $P(y|X,g)=P(y|X)$ assumption.
>
> * > This could be a better assumption and it seems that we could identify the subgroup information.
>
> Agreed! A benefit of the assumption $P(y|Z,g)=P(y|Z)$ is that we can now characterize an non-robust model's failure to capture the true distribution in terms of the loss gap between majority and minority groups (analogous to the sufficiency gap in [3]), This allows us to lay out a set of theoretical assumptions and derive their impact to the group identification performance (see next).

---

> ### Author Response · Authors · 2022-11-16
> **Response to Reviewer (Part 3)**
>
> * > Overall, I do think the paper requires clear theoretical assumptions and illustrates why the cross-validation based approach could identify the subgroup.
>
> Thanks for suggesting this additional theoretical contribution. We have derived a theoretical bound for the group identification performance of the proposed estimator based on generalization loss (included in the Appendix H of the updated paper).
>
> Indeed, after adopting the new assumption $P(y|Z,g)=P(y|Z)$, we can now characterize the behavior of a non-robust model (which fails to learn the true distribution) in terms of the difference in predictive distributions between the majority and the minority groups $P(y|\hat{y}=t,g=0) \neq P(y|\hat{y}=t, g=1)$. This implies a gap in the generalization error between the majority and the minority group $E(L(y, \hat{y}) |g=0) \neq E(L(y, \hat{y}) |g=1)$ under an unfair model.
>
> As a result, if we estimate the group membership using the rank of the cross-validation loss, (i.e., $1_{F(l_i) > q}$ where F is the CDF of model losses $l$ and q a user-specified percentile threshold), we can derive a data-dependent guarantee in the detection performance $P(g_i=1|F(l_i) > q)$ in terms of the majority-group prevalence $\gamma_0$, user-specified threshold $q$, and difference in the distribution of the majority- and minority-group losses $l_0$ and $l_1$ (Theorems H.1). This provides us a concrete understanding of how the  conditions in estimator configuration ($q$), data distribution ($\gamma_0$), and base model non-robust (i.e., difference in loss distribution) interact to impact the classifier performance. We have included this result in Appendix H and linked to it in Section 2.2.
>
> We would like to highlight that, to our knowledge, this is the first data-dependent bound for group detection performance in the debiasing literature (e.g., LfF, JTT, ARL, EIIL). The derivation of our guarantee is also non-trivial, as it requires developing novel bounds for the variances of empirical CDFs from first principles. Therefore, we believe this new result constitutes an additional contribution for the field.
>
> * > "I am not sure if the proposed approach is also ad-hoc or not."
>
> Following the previous discussion, we see that the identifiability of the procedure indeed relies on valid estimation of model generalization error (with respect to the true label). To this end, the K-fold cross-validation procedure we used in this work is known to produce unbiased and low-variance estimates for the expected generalization loss ([Blum et al 1999](https://www.ri.cmu.edu/pub_files/pub1/blum_a_1999_1/blum_a_1999_1.pdf), [Kumar et al, 2013](https://proceedings.mlr.press/v28/kumar13a.html)).
>
> 4. Practical Scalability
>
> * > I have doubts on the scalability of the proposed approach, because the bootstrapping approach would be quite long in a large scale dataset such as a language dataset. What is the time complexity of that?
>
> The complexity of the cross-validated ensemble approach (i.e., the cross-validated self-play) is identical to that of a single model training on full data (in the big O sense). This is because for cross-validated ensembles, each ensemble member model is trained on different subsets of the data. Therefore,
> If the complexity for a single model is T, then the complexity of the cross-validated ensemble is K*p*T, where p < 1 is the subsample percentage, and K is the ensemble size (both constants are under the control of the practitioner) . Therefore, in asymptotic notation, K*p*T=O(T) since p and K are both constants that don't scale with data size.
>
> Furthermore, in practice, p * K can be small (e.g, equal or not much greater than 1) depending on the configuration of the ensemble. For example, in our experiments, we find setting p=1/K (i.e., train K model on K non-overlap data splits) works rather well, and has the benefit of improving the cross-validation error estimate by reducing cross-split correlation [Blum et al, 1999](https://www.ri.cmu.edu/pub_files/pub1/blum_a_1999_1/blum_a_1999_1.pdf). This leads to p*K=1 and the complexity of cross-validated ensembles to be equal to T, (i.e., that of a single model on the full data). Furthermore, since ensemble training is embarrassingly parallel, the complexity can also be massively reduced leveraging multi-core machines to perform parallel training.
>
> We have expanded Appendix B.3 to include this discussion.
>
> * > It seems that we need multiple fine-tunings?
>
> Within each active sampling round, ISP does not involve multiple fine-tunings. Specifically, the full ISP procedure only conducts two rounds of training (i.e., the cross-validated ensemble training in Stage I, and standard training in Stage II). Both are done from scratch to maintain the simplicity of the method). It does not perform additional training otherwise.

---

> ### Comment · Reviewer_1jcC · 2022-11-16
> **Post-Rebuttal**
>
> Dear authors,
>
> Thank you for your thorough responses! The majority of my concerns have been addressed. Several minor points are as follows:
>
> After further consideration, I do think the invariant representation distribution is a correct assumption in your scenarios and is related to spurious correlation.
>
> Concerning uncertainty-based active learning in the context of a distribution shift. Given only a few data points in the specific subgroup, I would think the uncertainty estimation is still unreliable. By the way, a small amount of data in a distribution shift is always non-trivial, and the current paper makes several meaningful contributions to a better understanding of this problem. It would be greatly appreciated.
>
> Thanks again for the paper revision such as title modifications, fair notions. My final score will be updated after checking others reviews. No worries :).

---

> > ### Author Response · Authors · 2022-11-18
> > **Thanks and some additional discussions**
> >
> > Thanks for the supportive feedback! We are glad to see most of the concerns are addressed, and grateful to see your thoughtful comments about the value of this work :)
> >
> > You have raised a rather interesting point about the quality of uncertainty estimation for underrepresented groups, and we would like to provide some follow-up discussions regarding this:
> >
> > > Given only a few data points in the specific subgroup, I would think the uncertainty estimation is still unreliable.
> >
> > In machine learning modeling, there are usually two distinct types of uncertainties: **aleatoric uncertainty** and **epistemic uncertainty**  (see, e.g., [Hüllermeier and Waegeman (2021)](https://link.springer.com/article/10.1007/s10994-021-05946-3) for a recent review).
> > Briefly, the **aleatoric uncertainty** (or "data uncertainty") is caused by the external stochasticity inherent in the true data distribution (e.g., the variance $\sigma^2$ of observed data $y \sim (\mu, \sigma^2)$), while the **epistemic uncertainty** (or "model uncertainty") is the internal uncertainty in model estimation, caused by the lack of sufficient observed data to support confident parameter estimates (e.g., the Bayesian posterior uncertainty of mean estimates $Var( \mu  | y_1, \dots, y_n)$ ). The two sources of uncertainty are distinct from each other. For example, in a local region of the feature space when the number of observations $n \rightarrow \infty$, the aleatoric uncertainty estimate $E(\hat{\sigma}^2 | y_1, \dots, y_n)$ should converge to true $\sigma^2$, while the epistemic uncertainty $Var(\hat{\mu} | y_1, \dots, y_n)$ goes to 0. For our purpose, the **epistemic uncertainty** is of the most interest, since it informs us which types of examples are underrepresented in the feature space and needs additional sampling.
> >
> > To this end, the goal of ISP is to improve the model's ability in quantifying the **epistemic uncertainty** (i.e., to quantify whether there's sufficiently many examples similar to x in the training data) rather than the **data uncertainty** (i.e., estimating the distributional parameters of local distribution $p(y|x)$). Indeed, as the reviewer rightfully pointed out, when the number of examples is small, it is difficult to reliably estimate the aleatoric uncertainty (i.e., the precise values of the local distributional parameters). However, the epistemic uncertainty can still be properly estimated when the number of examples are small, as long as (1) the embedding distance correctly reflects the distance between the underrepresented v.s. the majority-group examples, and (2) we quantify epistemic uncertainty using a uncertainty measure that is based on embedding distance, e.g., the posterior distribution of a neural Gaussian process. (i.e., the _distance-aware principle_ for uncertainty quantification  [Liu et al 2020](https://papers.nips.cc/paper/2020/file/543e83748234f7cbab21aa0ade66565f-Paper.pdf), [van Amersfoort et al, 2021](https://arxiv.org/abs/2102.11409)). Here, the goal of ISP is exactly to improve epistemic uncertainty quality by improving representation.
> >
> > For example, in Figure 2 of the paper, even when there are only 2 minority examples per class (v.s. ~2500 majority group examples per class), the introspective training can properly quantify epistemic uncertainty for minority examples (i.e., returning high uncertainty in its neighborhood) due to its ability in separating the majority and minority examples in embedding space. Therefore, even when the number of underrepresented examples are few, we see introspective training still provides an improvement over ERM training by encouraging better embedding behavior. Of course, it would be theoretically very interesting to thoroughly investigate in what data situation would this advantage break down (esp. in high dimensional situations). However that is a bit out of scope of this current conference paper, and we will leave it for future work.

---

> > > ### Comment · Reviewer_1jcC · 2022-11-22
> > > **Thanks**
> > >
> > > Thanks for your explanations! Review has been updated. Best of luck!

---

### Author Response · Authors · 2022-11-18
**Summary of Reviewer Responses**

We thank all reviewers for your insightful comments and helpful suggestions.

We are glad to see the reviewers finding our work interesting [[dJid](https://openreview.net/forum?id=MofT9KEF0kw&noteId=xpGGqlHHxTS)], well-motivated [[1jcC](https://openreview.net/forum?id=MofT9KEF0kw&noteId=73K6JQprez)], clearly-written and well organized [[zGVn](https://openreview.net/forum?id=MofT9KEF0kw&noteId=wAQOwKty6x)], and that it proposed a meaningful solution for a difficult setting motivated by  real-world practice [[1jcC](https://openreview.net/forum?id=MofT9KEF0kw&noteId=73K6JQprez)]. We also appreciate the reviewers recognizing our theoretical contributions[[1jcC](https://openreview.net/forum?id=MofT9KEF0kw&noteId=73K6JQprez), [dJid](https://openreview.net/forum?id=MofT9KEF0kw&noteId=xpGGqlHHxTS)] and empirical evaluation [[1jcC](https://openreview.net/forum?id=MofT9KEF0kw&noteId=73K6JQprez), [zGVn](https://openreview.net/forum?id=MofT9KEF0kw&noteId=wAQOwKty6x), [dJid](https://openreview.net/forum?id=MofT9KEF0kw&noteId=xpGGqlHHxTS)].

Following the helpful suggestions from the reviewers, we have updated the manuscript with more accurate framing, additional theoretical contribution, and clarifications and discussions making the content more appropriate and accessible. The new content (in both main text and supplementary materials)  is highlighted in blue. We will work on adjusting the manuscript back to the appropriate number of pages post-rebuttal and when preparing camera-ready.

1. Improved positioning
   * We replaced the term "fairness" with "group robustness" throughout the paper to better highlight the core contribution. [[1jcC](https://openreview.net/forum?id=MofT9KEF0kw&noteId=73K6JQprez)]
   * We added a section In Appendix A.5 to discuss the connection between group robustness and fairness. [[1jcC](https://openreview.net/forum?id=MofT9KEF0kw&noteId=73K6JQprez), [zGVn](https://openreview.net/forum?id=MofT9KEF0kw&noteId=wAQOwKty6x)]
2. New theoretical contributions
   * In Section 1 "Notation and Problem Setup." and Section A.1, we improved the data assumption from $P(y|X)=P(y|X, g)$ to $P(y|Z)=P(y|Z,g)$. [[1jcC](https://openreview.net/forum?id=MofT9KEF0kw&noteId=73K6JQprez)]
   * In Appendix H, we derived a new data-dependent, bound on group detection performance (with very mild assumptions). It clarifies the impact of various conditions (estimator configuration, data distribution, model performance) on the estimator performance, and justifies the use of cross-validation estimators. [[1jcC](https://openreview.net/forum?id=MofT9KEF0kw&noteId=73K6JQprez)]
3. Additional method discussion and related work
   * In Appendix B.2., we added a discussion of the role of subgroup detection in the effectiveness of the overall ISP procedure. [[1jcC](https://openreview.net/forum?id=MofT9KEF0kw&noteId=73K6JQprez)]
   * In Appendix B.3, we added additional discussion of the computational complexity of the cross-validated ensemble procedure. [[1jcC](https://openreview.net/forum?id=MofT9KEF0kw&noteId=73K6JQprez)]
   * In Appendix D.1-3, we updated the description about IRM / EIIL using language suggested by reviewer [[1jcC](https://openreview.net/forum?id=MofT9KEF0kw&noteId=73K6JQprez)].
   * In Appendix D.2 (representation learning), we updated the description to stress the role of predefined fair notions in fair representation learning  [[1jcC](https://openreview.net/forum?id=MofT9KEF0kw&noteId=73K6JQprez)].
   * In Appendix D.4, we included recent works on active learning for distributional shift  [[1jcC](https://openreview.net/forum?id=MofT9KEF0kw&noteId=73K6JQprez)].
4. Updated Experiments
   * In Section 4, we updated the worst-group accuracy results for census income (Table 2). [[zGVn](https://openreview.net/forum?id=MofT9KEF0kw&noteId=wAQOwKty6x)].
   * In Section 4, we included DRO baseline results for census income and toxic comment  (Table 2). [[dJid](https://openreview.net/forum?id=MofT9KEF0kw&noteId=xpGGqlHHxTS)]
5. Updated Conclusion and Future Work
   * In Section 5, we suggested investigating performance wrt the choice of loss function / batch size as future work [[dJid](https://openreview.net/forum?id=MofT9KEF0kw&noteId=xpGGqlHHxTS)]

In addition, we respond to all clarifications requested by the reviewers individually.

---

### Decision · Program_Chairs · 2023-01-20

**Decision:**

Accept: poster

**Justification For Why Not Higher Score:**

There is some debate about the assumptions made in this paper and there is a scalability discussion in the current version.

**Justification For Why Not Lower Score:**

The paper is well written. The problem the paper focuses on is relevant, although the reviewers had concerns about the wording of fairness (the authors made revisions per reviewers' suggestions). The solution proposed in this paper is quite novel. There were some discussions about assumptions, but the authors adopted the suggestions from the reviewers.

**Metareview: Summary, Strengths And Weaknesses:**

This paper aims to improve the frontier of accuracy, which is the average performance across subgroups,  and group robustness, which is the worst performance across subgroups.  An active learning strategy is proposed to query the subgroup with worst performance to improve the accuracy and group robustness boundary.  This paper assumes that subgroup information is hidden and designs a novel ensemble based approach to detect/estimate the subgroup information. Both theoretical analyses and empirical investigations are demonstrated in this paper.

The paper is well written. The problem the paper focuses on is relevant, although the reviewers had concerns about the wording of fairness (the authors made revisions per reviewers' suggestions). The solution proposed in this paper is quite novel. There were some discussions about assumptions, but the authors adopted the suggestions from the reviewers.
Overall, the committee recommends acceptance of this paper.



**Note From Pc:**

if the above contains the word "oral" or "spotlight" please see: "oral" presentation means -> notable-top-5% and "spotlight" means -> notable-top-25%. As stated in our emails, we are disassociating presentation type from AC recommendations